# DECOUPLED PRIORITIZED RESAMPLING: ADVANCING OFFLINE RL WITH IMPROVED BEHAVIOR POLICY

## ABSTRACT

Offline reinforcement learning (RL) is challenged by the distributional shift problem. To tackle this issue, existing works mainly focus on designing sophisticated policy constraints between the learned policy and the behavior policy. However, these constraints are applied equally to well-performing and inferior actions through uniform sampling, which might negatively affect the learned policy. In this paper, we propose *Offline Decoupled Prioritized Resampling* (ODPR), which designs specialized priority functions for the suboptimal policy constraint issue in offline RL, and employs unique decoupled resampling for training stability. Through theoretical analysis, we show that the distinctive priority functions induce a provable improved behavior policy by modifying the distribution of the original behavior policy, and when constrained to this improved policy, a policy-constrained offline RL algorithm is likely to yield a better solution. We provide two practical implementations to balance computation and performance: one estimates priorities based on a fitted value network (ODPR-A), and the other utilizes trajectory returns (ODPR-R) for quick computation. ODPR serves as a highly compatible plug-and-play component with prevalent offline RL algorithms. We assess ODPR using five algorithms, namely BC, TD3+BC, Onestep RL, CQL, and IQL. Comprehensive experiments substantiate that both ODPR-A and ODPR-R markedly enhance the performance across all baseline methods.

## 1 INTRODUCTION

Offline Reinforcement Learning (RL) aims to solve the problem of learning from previously collected data without real-time interactions with the environment (Lange et al., 2012). However, standard off-policy RL algorithms tend to perform poorly in the offline setting due to the distributional shift problem (Fujimoto et al., 2019). Specifically, to train a Q-value function based on the Bellman optimality equation, these methods frequently query the value of out-of-distribution (OOD) state-action pairs, which leads to accumulative extrapolation error. Most existing algorithms tackle this issue by constraining the learning policy to stay close to the behavior policy that generates the dataset. These constraints directly operate on the policy densities, such as KL divergence (Jaques et al., 2019; Peng et al., 2019; Wu et al., 2019), Wasserstein distance (Wu et al., 2019), maximum mean discrepancy (MMD) (Kumar et al., 2019), and behavior cloning (Fujimoto & Gu, 2021).

However, such constraints might be too restrictive as the learned policy is forced to equally mimic bad and good actions of the behavior policy, especially in an offline scenario where data are generated by policies with different levels. For instance, consider a dataset $\mathcal{D}$ with state space $\mathcal{S}$ and action space $\mathcal{A} = \{\boldsymbol{a}_1, \boldsymbol{a}_2, \boldsymbol{a}_3\}$ collected with behavior policy $\beta$. At one specific state $\boldsymbol{s}^*$, the policy $\beta$ assigns probability 0.2 to action $\boldsymbol{a}_1$, 0.8 to $\boldsymbol{a}_2$ and zero to $\boldsymbol{a}_3$. However, $\boldsymbol{a}_1$ would lead to a much higher expected return than $\boldsymbol{a}_2$. Minimizing the distribution distance of two policies can avoid $\boldsymbol{a}_3$, but forces the learned policy to choose $\boldsymbol{a}_2$ over $\boldsymbol{a}_1$, resulting in much worse performance. Employing a policy constraint strategy is typically essential to avoid out-of-distribution actions. However, this necessity often results in a compromise on performance, stemming from a suboptimal policy constraint. Then the question arises: can we substitute the *behavior policy* with an improved one, enabling the learned policy to *avoid out-of-distribution actions and achieve superior policy constraint simultaneously*, thereby improving performance? Indeed, as illustrated in Figure 1, if we can accurately assess the quality of an action, we can then adjust its density to yield an improved behavior policy (blue).

Based on the above motivation, we propose data prioritization strategies for offline RL, *i.e.*, *Offline Decoupled Prioritized Resampling (ODPR)*. ODPR prioritizes data by the action quality, specifically,

assigning priority weight proportional to normalized (*i.e.* non-negative) advantage — the additional reward that can be obtained from taking a specific action. In practice, we develop two implementations, *Advantage-based ODPR (ODPR-A)* and *Return-based ODPR (ODPR-R)*. ODPR-A fits a value network from the dataset and calculates advantages with one-step TD error for all transitions. We further advance ODPR-A by iteratively refining current behavior policy based on the previous one. Similarly, ODPR-R, a more computation-friendly version, employs trajectory return as the priority when trajectory information is available. Then, both ODPR-A and ODPR-R run an offline RL algorithm with the prioritized behavior policy to learn a policy. Our contributions are three-fold:

- **Innovative Approaches to Prioritized Replay for Offline RL:** We introduce a unique class of priority functions specifically tailored for offline RL. Contrary to online prioritized resampling methods like PER (Schaul et al., 2016), which mainly aim to accelerate value function fitting, our proposed priority is motivated by the desire to cultivate a superior behavior policy. We further enhance the improved behavior policy through iterative prioritization. Another notable aspect of our methodology, distinguishing it from existing resampling methods in both online and offline RL, is the incorporation of dual samplers: a uniform one for policy evaluation and a prioritized one for policy improvement and constraint, referred to as *decoupled sampling*. We demonstrate that decoupled sampling is crucial for maintaining the stability of offline resampling training.

- **Theoretical Improvement Guarantee:** We theoretically demonstrate that a prioritized behavior policy, with our proposed priority functions, yields a higher expected return than the original one. Furthermore, under some special cases, we theoretically show that a policy-constrained offline RL problem has an improved optimal solution when the behavior policy is prioritized.

- **Empirical Compatibility and Effectiveness:** We conduct extensive experiments to reveal that our proposed prioritization strategies boost the performance of prevalent offline RL algorithms across diverse domains in D4RL (Brockman et al., 2016; Fu et al., 2020). The performance of CQL, IQL, and TD3+BC has been improved significantly by 34, 46, and 67 points, respectively, on the Mujoco locomotion tasks, which shows ODPR is a plug-in orthogonal to algorithmic improvements. Furthermore, we demonstrate that, owing to its unique *fine-grained priorities*, ODPR-A is applicable within datasets without complete trajectories, a scenario where preceding trajectory-based resampling strategies in offline RL have been unsuccessful.

## 2 PRELIMINARIES

**Reinforcement Learning (RL).** RL addresses the problem of sequential decision-making, which is formulated with a Markov Decision Process $\langle \mathcal{S}, \mathcal{A}, T, r, \gamma \rangle$. Here, $\mathcal{S}$ is a finite set of states; $\mathcal{A}$ is the action space; $T(\boldsymbol{s}, \boldsymbol{a}, \boldsymbol{s}')$ is the dynamics function; $r(\boldsymbol{s}, \boldsymbol{a})$ and $\gamma \in (0, 1]$ are the reward function and the discount factor respectively. The policy is denoted by $\pi(\boldsymbol{a}|\boldsymbol{s})$ and an induced trajectory is denoted by $\tau$. The goal of RL is to learn a policy maximizing the expected cumulative discounted reward:

$$J(\pi) = \mathbb{E}_{\tau \sim p_\pi(\tau)} \left[ \sum_{t=0}^{\infty} \gamma^t r(\boldsymbol{s}_t, \boldsymbol{a}_t) \right]. \quad (1)$$

**Offline RL as Constrained Optimization.** Offline RL considers a dataset $\mathcal{D}$ generated with behavior policy $\beta$. Since $\beta$ or $\mathcal{D}$ is fixed throughout training, maximizing $J(\pi)$ is equivalent to maximizing the improvement $J(\pi) - J(\beta)$. It can be measured by:

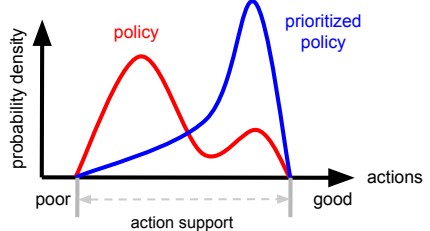

Figure 1: Action Prioritization. Actions in x-axis are ranked by their quality. A behavior policy (in red) usually follows a multimodal distribution. A prioritized policy (in blue) modifies the policy densities by assigning higher weights to better actions. The two policies share the same action support (action coverage).

**Lemma 2.1.** (Performance Difference Lemma (Kakade & Langford, 2002).) *For any policy $\pi$ and $\beta$,*

$$J(\pi) - J(\beta) = \int_{\boldsymbol{s}} d_\pi(\boldsymbol{s}) \int_{\boldsymbol{a}} \pi(\boldsymbol{a}|\boldsymbol{s}) A^\beta(\boldsymbol{s}, \boldsymbol{a}) \, d\boldsymbol{a} \, d\boldsymbol{s}, \quad (2)$$

*where $d_\pi(\boldsymbol{s}) = \sum_{t=0}^{\infty} \gamma^t p(\boldsymbol{s}_t = \boldsymbol{s}|\pi)$, represents the unnormalized discounted state marginal distribution induced by the policy $\pi$, and $p(\boldsymbol{s}_t = \boldsymbol{s}|\pi)$ is the probability of the state $\boldsymbol{s}_t$ being $\boldsymbol{s}$.*

The proof can be found in Appendix B.1. We consider the offline RL paradigm with policy constraint, which enforces the policy $\pi$ to stay close to the behavior policy $\beta$. Therefore, following TRPO (Schul-

man et al., 2015), Equation (2) can be approximated as $\hat{\eta}(\pi, \beta) \approx \int_{\boldsymbol{s}} d_\beta(\boldsymbol{s}) \int_{\boldsymbol{a}} \pi(\boldsymbol{a}|\boldsymbol{s}) A^\beta(\boldsymbol{s}, \boldsymbol{a}) \, d\boldsymbol{a} \, d\boldsymbol{s}$. Hence, $\hat{\eta}(\pi, \beta)$ represents the performance improvement of $\pi$ over $\beta$. Offline RL is to maximize $J(\pi)$ while constraining $\pi$ to be closed to $\beta$. It can be formulated as the following constrained optimization problem with an expected KL-divergence constraint:

$$\pi^* = \arg\max_\pi \hat{\eta}(\pi, \beta) \quad \text{s.t.} \quad \int_{\boldsymbol{s}} d_\beta(\boldsymbol{s}) \mathrm{D}_{\mathrm{KL}}\left(\pi(\cdot|\boldsymbol{s})||\beta(\cdot|\boldsymbol{s})\right) d\boldsymbol{s} \leq \epsilon \quad \text{and} \quad \int_{\boldsymbol{a}} \pi(\boldsymbol{a}|\boldsymbol{s}) \, d\boldsymbol{a} = 1. \quad (3)$$

An analytic solution $\pi^*$ of the above problem is given by Peng et al., 2019 (see Appendix B.2). Related works on offline RL and prioritized sampling are in Appendix A due to space constraints.

## 3 Offline Decoupled Prioritized Resampling

In this section, we develop Offline Decoupled Prioritized Resampling, which prioritizes transitions in an offline dataset at training according to a class of priority functions. We start with an observation that performing prioritized sampling on a dataset generated with policy $\beta$ is equivalent to sampling from a new behavior $\beta'$. Then, we theoretically justify that $\beta'$ gives better performance than $\beta$ in terms of the cumulative return when proper priority functions are chosen. In the end, we propose two practical implementations of ODPR using transition advantage and return as the priority, respectively.

### 3.1 Prioritized Behavior Policy

Consider a dataset $\mathcal{D}$ generated with behavior policy $\beta$. Let $\omega(\boldsymbol{s}, \boldsymbol{a})$ denote a weight/priority function for transitions in $\mathcal{D}$. Then, we define a prioritized behavior policy $\beta'$:

$$\beta'(\boldsymbol{a}|\boldsymbol{s}) = \frac{\omega(\boldsymbol{s}, \boldsymbol{a})\beta(\boldsymbol{a}|\boldsymbol{s})}{\int_{\boldsymbol{a}} \omega(\boldsymbol{s}, \boldsymbol{a})\beta(\boldsymbol{a}|\boldsymbol{s})d\boldsymbol{a}}, \quad (4)$$

where the denominator is to guarantee $\int_{\boldsymbol{a}} \beta'(\boldsymbol{a}|\boldsymbol{s}) \, d\boldsymbol{a} = 1$. As shown in Figure 1, $\beta'$ shares the same action support as $\beta$. Suppose a dataset produced by prioritized sampling on $\mathcal{D}$ is $\mathcal{D}'$. We have:

$$\mathbb{E}_{(\boldsymbol{s}, \boldsymbol{a}) \sim \mathcal{D}'}\left[\mathcal{L}_\theta(\boldsymbol{s}, \boldsymbol{a})\right] = \mathbb{E}_{\boldsymbol{s} \sim \mathcal{D}, \boldsymbol{a} \sim \beta'(\cdot|\boldsymbol{s})}\left[\mathcal{L}_\theta(\boldsymbol{s}, \boldsymbol{a})\right], \quad (5)$$

where $\mathcal{L}$ represents a generic loss function, and the constant is discarded as it does not affect the optimization. This equation shows that prioritizing the transitions in a dataset by resampling or reweighting (LHS) can mimic the behavior of another policy $\beta'$ (RHS).

Intuitively, priority functions, denoted as $\omega(\boldsymbol{s}, \boldsymbol{a})$, should be *non-negative* and *monotonically increase with respect to the quality of the action* $\boldsymbol{a}$. In the context of RL, advantage $A^\beta(\boldsymbol{s}, \boldsymbol{a})$ represents the extra reward that could be obtained by taking the action $\boldsymbol{a}$ over the expected return. Therefore, advantage $A^\beta(\boldsymbol{s}, \boldsymbol{a})$, as an action quality indicator, provides a perfect tool to construct $\omega(\boldsymbol{s}, \boldsymbol{a})$. We can easily construct many functions that satisfy the above properties, such as a linear function

$$\omega(A^\beta(\boldsymbol{s}, \boldsymbol{a})) = C(A^\beta(\boldsymbol{s}, \boldsymbol{a}) - \min_{(\boldsymbol{s}, \boldsymbol{a}) \in \mathcal{D}} A^\beta(\boldsymbol{s}, \boldsymbol{a})), \quad (6)$$

where $C$ is a constant, set to make the sum over the dataset equal to 1.

### 3.2 Prioritized Policy Improvement

We are ready to show that prioritized sampling can contribute to an improved *learned policy*. We first show the prioritized version $\beta'$ is superior to $\beta$.

**Behavior Policy Improvement.** The below theorem underscores that prioritization can improve the original behavior policy $\beta$ if it is a stochastic policy or a mixture of policies, either of which could result in actions of different Q-values. The detailed proof is deferred to Appendix B.3.

**Theorem 3.1.** *Let $\omega(A)$ be any priority function with non-negative and monotonic increasing properties. Then, we have $J(\beta') - J(\beta) \geq 0$. If there exists a state $\boldsymbol{s}$, under which not all actions in action support $\{\boldsymbol{a}|\beta(\boldsymbol{a}|\boldsymbol{s}) > 0, \boldsymbol{a} \in \mathcal{A}\}$ have the same Q-value, the inequation strictly holds.*

**Learned Policy Improvement.** Further, under two special cases, we establish an improvement guarantee on the learned policy. Consider the constrained optimization problem defined by Equation (3), we use $\pi^*$ and $\pi'^*$ to denote the optimal solution regarding behavior $\beta$ and $\beta'$ respectively. Our expectation is that $\pi'^*$ is better than $\pi^*$ in terms of cumulative return, *i.e.*, $J(\pi'^*) \geq J(\pi^*)$. In an extreme case, if the policy constraint is exceptionally strong, causing the learned policy to

exhibit performance very similar to the behavior policy, $\pi'^*$ obviously surpasses $\pi^*$ because $\beta'$ is greater than $\beta$. In another more general case with a certain KL-divergence policy constraint, we show that if the state marginal distribution induced by $\beta'$ is close to the distribution induced by $\beta$, the learned policy $\pi'^*$ can be surely improved over $\pi^*$. To show this, we use the cumulative return of $\beta$ as a baseline to compare the performance differences $\hat{\eta}(\pi'^*, \beta)$ and $\hat{\eta}(\pi^*, \beta)$. Formally, when we assume $d_{\beta'}(\boldsymbol{s}) = d_\beta(\boldsymbol{s})$, using $\omega(A^\beta(\boldsymbol{s}, \boldsymbol{a}))$ defined in Equation (6), we have $\hat{\eta}(\pi'^*, \beta) \geq \hat{\eta}(\pi^*, \beta)$, where $\pi^*$ defined by Equation (3), $\pi'^*$ is defined as $\pi'^* = \arg\max_{\pi'} \hat{\eta}(\pi', \beta)$, s.t. $\int_{\boldsymbol{s}} d_{\beta'}(\boldsymbol{s}) \mathrm{D}_{\mathrm{KL}}\left(\pi'(\cdot|\boldsymbol{s})||\beta'(\cdot|\boldsymbol{s})\right) d\boldsymbol{s} \leq \epsilon$, and $\int_{\boldsymbol{a}} \pi'(\boldsymbol{a}|\boldsymbol{s}) d\boldsymbol{a} = 1, \quad \forall\, \boldsymbol{s}$. The inequation strictly holds under the same condition with Theorem 3.1. See Appendix B.4 for detailed proof.

In this way, we have $J(\pi'^*) \geq J(\pi^*)$, which demonstrates that $\pi'^*$ is a better solution. Although with an assumption about state distribution, it still offers valuable insights that the constraint induced by prioritized behavior policy has the potential to improve the performance of the learned policy. The rationale behind this is straightforward: when starting from a better behavior policy (Theorem 3.1), the learned policy is more likely, though not guaranteed, to achieve a higher final performance.

**Connections to policy-constrained offline RL algorithms.** In offline RL, many methods fall into this KL-constrained framework. IQL, AWAC, CRR, and OnestepRL extract policy by exponential advantage regression, induced from KL divergence (Peng et al., 2019). Kostrikov et al. (2021a) shows that CQL can be viewed as a KL divergence regularization between the Boltzmann policy and the behavior policy. The BC term in TD3+BC is exactly KL divergence under Gaussian policy with fixed variance. Therefore, our analysis should be applicable to these algorithms.

## 3.3 PRACTICAL ALGORITHMS

**Advantage-based Offline Decoupled Prioritized Resampling (ODPR-A).** We approximate priorities by fitting a value function $V_\psi^\beta(\boldsymbol{s})$ for the behavior policy $\beta$ by TD-learning:

$$\min_\psi \quad \mathbb{E}_{(\boldsymbol{s}, \boldsymbol{a}, \boldsymbol{s}', r) \sim \mathcal{D}}\left[\left(r + \gamma V_\psi(\boldsymbol{s}') - V_\psi(\boldsymbol{s})\right)^2\right]. \tag{7}$$

The advantage for $i$-th transition $(\boldsymbol{s}_i, \boldsymbol{a}_i, \boldsymbol{s}_i', r_i)$ in the dataset is then given by a one-step TD error:

$$A(\boldsymbol{s}_i, \boldsymbol{a}_i) = r_i + V_\psi(\boldsymbol{s}_i') - V_\psi(\boldsymbol{s}_i), \tag{8}$$

which is similar to the form of priority in online PER, such as the absolute TD error, but differs in whether the absolute value is taken. This implementation is referred to as *ODPR-A* in the following. The term "decoupled" will be elucidated subsequently within this section.

**Return-based Offline Decoupled Prioritized Resampling (ODPR-R).** The limitation of ODPR-A is also clear, *i.e.*, fitting the value network incurs extra computational cost. Therefore, we propose another variant that uses trajectory return as an alternative transition quality indicator. For the $i$-th transition, we find the complete trajectory that contains it, and calculate the return for the whole trajectory $G_i = \sum_{k=0}^{T_i} r_k$. $T_i$ is the length of the trajectory. Then the priority is obtained by

$$\omega_i = C\left(\frac{G_i - G_{\min}}{G_{\max} - G_{\min}} + p_{\mathrm{base}}\right), \tag{9}$$

where $G_{\min} = \min_i G_i$ and $G_{\max} = \max_i G_i$. $p_{\mathrm{base}}$ is a small positive constant that prevents zero weight. $C$ is a constant, set to make the sum equal to 1. We term this variant as *ODPR-R*. ODPR-R can only work with datasets where the trajectory information is available. We compare the characteristics of ODPR-A and ODPR-R in Table 1.

**Decoupled prioritized resampling.** After obtaining priority weights, ODPR can be implemented by both resampling and reweighting. The sampling probability or weight is proportional to its priority. We opt for resampling in the main text and also provide the results of reweighting in the Appendix D.7. An offline RL algorithm can be decomposed into three components:

Table 1: A summary for two algorithms.

|  | ODPR-A | ODPR-R |
|---|---|---|
| Prerequisite | None | full trajectory |
| Extra Runtime | fit value function | $< 3$ seconds |
| Feature | weights can be reused | |

policy evaluation, policy improvement, and policy constraint. In alignment with our stated motivation, we employ prioritized data for both policy constraint and policy improvement terms to mimic being constrained to a better behavior policy. However, we found it is crucial to conduct policy evaluation under non-prioritized data for stability. Such a prioritization method is termed as *decoupled*

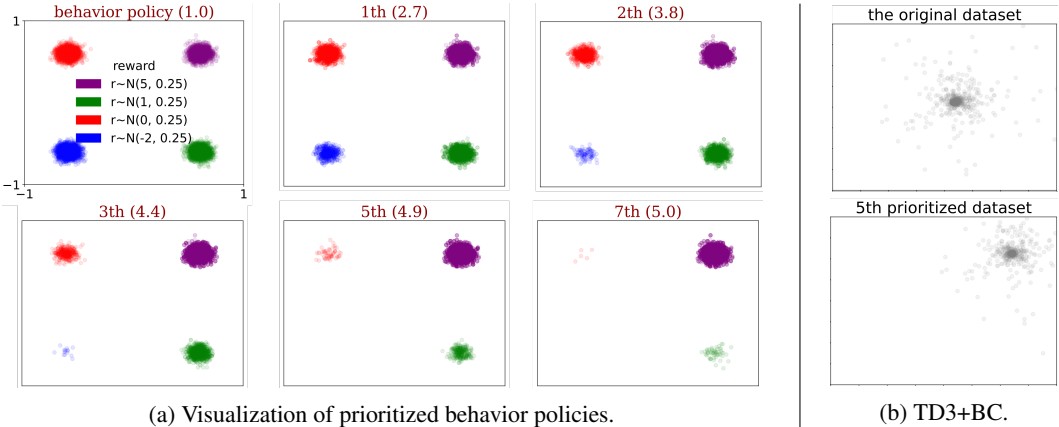

(a) Visualization of prioritized behavior policies.

(b) TD3+BC.

Figure 2: An illustration of the effect of ODPR on bandit. (a) Visualization of prioritized behavior policies. As iteration goes, the prioritized dataset gradually converges to the optimal action mode (purple). The value in parentheses represents the average reward. (b) The upper figure represents TD3+BC learning on the original dataset, which failed to find the optimal action. In contrast, the lower figure represents TD3+BC on the 5th prioritized dataset, converging to the optimal mode.

*prioritized resampling*. For decoupled prioritized resampling, two samplers are employed, one for uniform sampling and one for prioritized sampling. A more in-depth discussion about why decoupled resampling is crucial can be found in Section 4.3. The complete algorithm is given in Algorithm 1.

---

**Algorithm 1** Offline Decoupled Prioritized Resampling

---

1: **Require:** Dataset $\mathcal{D} = \{(\boldsymbol{s}, \boldsymbol{a}, \boldsymbol{s}', r)_i\}_{i=1}^{N}$, a policy-constrained algorithm $\mathcal{I}$
2: **Stage1:** Calculate $\omega_i$ according to Equation (8) or Equation (9) ( with trajectory information).
3: **Stage2 (Decoupled Resampling):** Train algorithm $\mathcal{I}$ on dataset $\mathcal{D}$. Sample transition $i$ with the priority $\omega_i$ for policy constraint and improvement. Uniform sample for policy evaluation.

---

### 3.4 Improving ODPR-A by Iterative Prioritization

In Section 3.2, we demonstrate a likelihood that enhancing $\beta(\boldsymbol{a}|\boldsymbol{s})$ to $\beta'(\boldsymbol{a}|\boldsymbol{s})$ leads to an improvement in the learned policy through offline RL algorithms. Then, a natural question arises: can we further boost the learned policy by improving $\beta'(\boldsymbol{a}|\boldsymbol{s})$? The answer is yes. Suppose we have a sequence of behavior policies $\beta^{(0)}, \beta^{(1)}, \ldots, \beta^{(K)}$ satisfying $\beta^{(k)}(\boldsymbol{a}|\boldsymbol{s}) \propto \omega(A^{(k-1)}(\boldsymbol{a}, \boldsymbol{s}))\beta^{(k-1)}(\boldsymbol{a}|\boldsymbol{s})$, where $A^{(k-1)}(\boldsymbol{a}, \boldsymbol{s})$ represents the advantage for policy $\beta^{(k-1)}(\boldsymbol{a}|\boldsymbol{s})$. We can easily justify that the behavior policies are monotonically improving by Theorem 3.1:
$$J(\beta^{(0)}) \leq J(\beta^{(1)}) \leq J(\beta^{(2)}) \leq \cdots \leq J(\beta^{(K)}).$$

It is reasonable to anticipate, though not guarantee, the following relationship: $J(\pi^{(0)*}) \leq J(\pi^{(1)*}) \leq J(\pi^{(2)}) \leq \cdots \leq J(\pi^{(K)*})$, where $\pi^{(k)*}$ is the optimal solution of Equation (3) when constrained to $\beta^{(k)}$. We build such a sequence of behaviors from a fixed policy $\beta^{(0)} = \beta$ and its dataset $\mathcal{D}$, which relies on the recursion $\beta^{(k)}(\boldsymbol{a}|\boldsymbol{s}) \propto \prod_{j=0}^{k-1} \omega(A^{(j)}(\boldsymbol{a}, \boldsymbol{s})) \cdot \beta^{(0)}(\boldsymbol{a}|\boldsymbol{s})$.

It means that a dataset $\mathcal{D}^{(k)}$ for behavior $\beta^{(k)}$ can be acquired by resampling the dataset $\mathcal{D}$ with weight $\prod_{j=0}^{k-1} \omega(A^{(j)}(\boldsymbol{a}, \boldsymbol{s}))$ (normalize the sum to 1). Then, the advantage $A^{(k)}$ can be estimated on $\mathcal{D}^{(k)}$ following Equation (7)-Equation (8). After all iterations, we scale the standard deviation of priorities to a hyperparameter $\sigma$ to adjust the strength of data prioritization. The full algorithm for this iterative ODPR-A is presented in Algorithm 2. In the experiments, ODPR-A mainly refers to this improved version. It is notable that priorities that are acquired in the first stage can be saved and made public, and then offline RL algorithms could directly use the existing priorities without extra cost.

## 4 Experiments

We start with a simple bandit experiment to illustrate the effect of ODPR-A and ODPR-R. Then we apply our methods to the state-of-the-art offline RL algorithms to show their effectiveness on the D4RL benchmark. Further, we conduct experiments to analyze the essential components in ODPR.

Table 2: Normalized scores on MuJoCo locomotion v2 tasks. We report the average and the standard deviation (SD) of the total score over 15 seeds. The results that have an advantage over the baselines (denoted as vanilla) are printed in **bold** type. "m", "mr", and "me" are respectively the abbreviations for "medium", "medium-replay", and "medium-expert". "V", "A", and "R" denotes "vanilla", "ODPR-A ", and "ODPR-R ". Standard deviation of individual games can be found at Appendix D.5.

| Dataset | TD3+BC | | | CQL | | | IQL | | | OnestepRL | | |
|---|---|---|---|---|---|---|---|---|---|---|---|---|
| | V | A | R | V | A | R | V | A | R | V | A | R |
| halfcheetah-m | 48.3 | **50.0** | **48.6** | 48.2 | 48.3 | 48.1 | 47.6 | 47.5 | 47.6 | 48.4 | **48.6** | 48.4 |
| hopper-m | 57.3 | **74.1** | **59.1** | 72.1 | **72.7** | **74.9** | 64.1 | **66.0** | **66.4** | 57.2 | **64.8** | **58.2** |
| walker2d-m | 84.9 | 84.9 | 84.2 | 82.1 | **83.9** | 80.7 | 80.0 | **83.9** | 78.3 | 77.9 | **85.1** | **80.9** |
| halfcheetah-mr | 44.5 | **45.9** | 44.6 | 45.2 | 45.4 | **46.1** | 43.4 | 43.0 | **44.0** | 37.5 | **42.9** | **39.7** |
| hopper-mr | 58.0 | **88.7** | **77.4** | 96.1 | 94.2 | 92.3 | 88.4 | **95.3** | **99.9** | 90.1 | 82.6 | 90.6 |
| walker2d-mr | 72.9 | **88.2** | **82.7** | 82.3 | **85.9** | 81.7 | 69.1 | **82.7** | **79.1** | 58.2 | **72.4** | **63.7** |
| halfcheetah-me | 92.4 | 83.3 | **93.9** | 62.1 | **70.7** | **84.3** | 82.9 | **92.7** | **93.5** | 94.1 | 94.2 | 93.9 |
| hopper-me | 99.2 | **107.3** | **106.7** | 82.9 | **105.1** | 97.2 | 97.2 | **105.1** | **107.2** | 80.5 | **99.4** | **98.8** |
| walker2d-me | 110.2 | **111.7** | 110.1 | 110.0 | 107.9 | 109.6 | 109.4 | **111.6** | **110.7** | 111.1 | **112.5** | 111.4 |
| total | 667.7 | 734.1 | 707.3 | 681.0 | 714.1 | 714.9 | 682.1 | 727.8 | 726.7 | 655.0 | 702.5 | 685.6 |
| SD(total) | 18.4 | 10.4 | 7.9 | 15.3 | 6.2 | 14.9 | 22.3 | 11.2 | 8.9 | 21.7 | 6.2 | 16.7 |

## 4.1 TOY BANDIT PROBLEM

We consider a bandit task, where the action space is 2D continuous, $\mathcal{A} = [-1, 1]^2$ (Wang et al., 2023) and as a bandit has no states, the state space $\mathcal{S} = \emptyset$. The offline dataset is as the first figure in Figure 2a shows (see Appendix C.1 for details). The goal of the bandit task is to learn the action mode with the highest expected reward from the offline dataset. To demonstrate the effect of ODPR-A, We show that TD3+BC fails to find the optimal action, while with ODPR-A, it solves the problem.

We first show that prioritized datasets are improved over the original one in Figure 2a. The blue samples with the lowest reward are substantially reduced in the first two iterations. After iterating five times, the suboptimal red and green samples also significantly diminish. The average return of the prioritized dataset is increased to 4.9, very close to the value of optimal actions. In the 7th iteration, suboptimal actions almost disappear. Since the reward is exactly the return in bandit, ODPR-R is the 1st prioritized behavior policy of ODPR-A, which raises the average return from 1.0 to 2.69.

Next, we show how offline RL algorithms can be improved by ODPR-A. As Figure 2b shows, when trained on the original dataset, TD3+BC failed to produce the optimal policy since it is negatively affected by suboptimal actions and converges to $(0.2, 0.2)$, the mean of four modes (policy constraint) but biased towards the best action (policy improvement). However, if combined with ODPR-A (iteration K=5), it successfully finds the optimal mode.

## 4.2 D4RL BENCHMARK

In this section, experiments on D4RL benchmark are conducted to empirically show Offline Decoupled Prioritized Resampling can improve popular offline RL algorithms on diverse domains.

**Experiment Setups.** As discussed in Section 3.2, behavior cloning (BC), as a special case of offline RL, can be improved by ODPR. In addition, ODPR is a general plug-and-play training scheme that improves a variety of state-of-the-art (SOTA) offline RL algorithms. In our work, we choose four widely adopted algorithms as case studies, CQL, OnestepRL, IQL, and TD3+BC.

ODPR's priority weights are generated in the first stage and then can be reused among baselines and seeds, saving computation. However, to assess the variance of ODPR and verify the generalization ability of ODPR to different algorithms, we organize experiments by sharing priority weights among baselines but not seeds. Specifically, we take seed=1 to compute ODPR-A weights, and then apply these weights and seed=1 to run TD3+BC, IQL, *etc*.We subsequently repeat this process with the next random seed. More experiment settings can be found in Appendix C.2 and Appendix C.3.

**Mujoco locomotion.** Table 3a reveals that ODPR induces a better offline dataset, from which behavior cloning produces a behavior policy with higher performance. Further, Table 2 shows that even though the state-of-the-art algorithms have achieved a strong performance, ODPR-A and ODPR-R can further improve the performance of **all** algorithms by large margins. Specifically, with ODPR-A, TD3+BC achieves a total score of 734.1 from 667.7. In addition, IQL, when combined

with ODPR-A and ODPR-R, also reaches 727.8 and 726.7 points, respectively. We observe that ODPR-A generally performs better than ODPR-R. This is potentially because ODPR-A is improved by iterative prioritization while ODPR-R simply utilizes trajectory returns. Interestingly, ODPR occasionally attains a smaller standard deviation than the vanilla, mainly due to its ability to achieve higher and more stable scores in some difficult environments. Another notable observation is that although TD3+BC performs worse than IQL and CQL in their vanilla implementations, TD3+BC eventually obtains the highest performance boost with ODPR-A and achieves the best performance with a score of 734.1. The reason might be that TD3+BC directly constrains the policy with a BC term, which is easier to be affected by negative samples.

(a) Averaged normalized scores of Behavior Cloning (BC) on MuJoCo locomotion v2 tasks over 15 seeds.

| Dataset | BC | | |
|---|---|---|---|
| | V | A | R |
| halfcheetah-m | 42.7 | **46.5** | 42.6 |
| hopper-m | 48.3 | **57.4** | 52.2 |
| walker2d-m | 73.3 | **83.8** | 70.1 |
| halfcheetah-mr | 33.4 | **41.6** | 39.1 |
| hopper-mr | 31.1 | **56.1** | 30.3 |
| walker2d-mr | 26.5 | **81.2** | 48.2 |
| halfcheetah-me | 62.8 | **95.4** | 81.1 |
| hopper-me | 52.3 | **110.7** | 71.2 |
| walker2d-me | 106.4 | **110.9** | 107.4 |
| total | 476.8 | **683.6** | 542.2 |
| SD(total) | 17.7 | 8.1 | 18.2 |

(b) Averaged normalized scores on Antmaze, Kitchen, and Adroit tasks over 15 seeds. The results for AW (Hong et al., 2023) were taken directly from the official source[1].

| | | IQL | AW | ODPR-A | ODPR-R |
|---|---|---|---|---|---|
| antmaze | umaze | 88.5 | **90.7** | 85.5 | 87.8 |
| | umaze-diverse | 63.1 | **75.3** | 70.8 | 66.0 |
| | medium-play | 70.5 | 61.3 | **76.1** | 72.0 |
| | medium-diverse | 58.5 | 22.0 | **71.8** | 74.2 |
| | large-play | 44.1 | 23.3 | 40.0 | **49.6** |
| | large-diverse | 42.0 | 9.3 | **48.0** | 43.0 |
| | antmaze total | 366.7 | 281.9 | 392.2 | **392.6** |
| kitchen | complete-v0 | 65.9 | 26.3 | 64.2 | 62.7 |
| | partial-v0 | 51.4 | **73.1** | 66.5 | 69.5 |
| | mixed-v0 | 50.3 | 47.8 | 52.1 | 49.9 |
| | kitchen total | 167.6 | 147.2 | **182.8** | 182.1 |
| pen | human-v0 | 73.1 | - | 72.9 | **83.0** |
| | cloned-v0 | 42.1 | - | 61.2 | **66.6** |

Table 3: Experiment results of ODPR.

**Discussions on Data Prioritizing.** In particular, we observe that on the locomotion tasks, the performance boost of ODPR-A and ODPR-R mainly comes from the "medium-replay" and "medium-expert" level environments. To better understand this phenomenon, we visualize trajectory return distributions of hopper on these two levels in Figure 6. The visualizations suggest that these tasks have a highly diverse data distribution. This is consistent with our intuition that the more diverse the data quality is, the more potential for the data to be improved through data prioritizing by quality. More experiments with varying behavior policies can be found at Appendix D.1 and Appendix D.2.

**Antmaze, Kitchen and Adroit.** In addition to the locomotion tasks, we evaluate our methods in more challenging environments. Given that IQL achieves the absolute SOTA performance in these domains and other algorithms, e.g., CQL, do not give an ideal performance in these domains, we pick IQL as a case study. We present the results in Table 3b. Similarly, we observe that both ODPR-A and ODPR-R can further improve the performance of IQL on all three domains. In the most challenging Antmaze environments, ODPR-A and ODPR-R successfully improve the most difficult medium and large environments. For Kitchen and Adroit tasks, we have observed a similar trend of improvement.

## 4.3 ABLATION STUDIES

**Effect of the number of iterations K.** In Table 4, as the iteration progresses, the overall performance of BC and TD3+BC combined with ODPR-R on the locomotion tasks continues to increase. For BC, performance declines when K = 5, which may be due to some transitions having dispro-

Table 4: Effect of the number of iterations $K$ (15 seeds).

| $K$ | vanilla | 1 | 2 | 3 | 4 | 5 |
|---|---|---|---|---|---|---|
| BC | 476.8 | 651.0 | 674.5 | 664.5 | **683.6** | 662.1 |
| TD3+BC | 667.7 | 711.2 | 706.3 | 719.7 | 725.1 | **734.1** |

portionately large or small weights, which affects the gradient descent's convergence. However, the improvement is significant even with just one iteration compared to the original algorithm. We typically choose a value of K between 3 and 5 for the best performance.

**Comparison with longer training.** ODPR-A requires additional computational cost to calculate priorities. To provide a fair comparison, we ran vanilla TD3+BC for twice as long. We found that

[1] https://github.com/Improbable-AI/harness-offline-rl

TD3+BC converges rapidly, and the results at 2M steps were similar to those at 1M steps (677.7 vs. 672.7). This indicates that the superior performance of ODPR-A is not due to extra computation, but rather stems from the improved policy constraint.

**Analysis of decoupled resampling.** A basic recipe of offline RL algorithms comprises policy evaluation, policy improvement, and policy constraint. Since ODPR focuses on producing a better behavior policy for policy constraint, it seems natural to solely apply prioritized data to the constraint term. However, this does not always improve performance. For instance, as Table 5 shows, on TD3+BC with ODPR-R, only prioritizing data for constraint results in a dramatic drop. We observed that it suffers from extrapolation error and results in Q-value overestimation in several tasks. We suspect it is because, when only prioritizing the constraint term, it imposes a weaker constraint on low-priority actions,

Table 5: Effect of decoupled resampling. Results are the total scores on Mujoco tasks with 15 seeds. "+CNT" and "+IPV" denotes using the prioritized sampling for policy constraint and policy improvement term, respectively[2]. "all" denotes using the prioritized sampling for all three terms. Very low scores are marked with red.

|        | vanilla |        | +CNT | +CNT+IPV | all |
|--------|---------|--------|------|----------|-----|
| TD3+BC | 667.7   | ODPR-A | 723.7 | **734.1** | **731.7** |
|        |         | ODPR-R | 608.7 | **707.6** | 707.3 |
| CQL    | 681.0   | ODPR-A | 674.9 | **714.1** | 652.8 |
|        |         | ODPR-R | 672.3 | **714.9** | 708.3 |
| IQL    | 682.1   | ODPR-A | -    | **727.8** | 674.3 |
|        |         | ODPR-R | -    | 706.5    | **726.7** |
| OnestepRL | 655.0 | ODPR-A | -    | **702.5** | 658.1 |
|        |         | ODPR-R | -    | **685.6** | 681.1 |

while the policy improvement remains unchanged. As a result, the extrapolation error of low-priority samples accumulates. To validate our hypothesis, if we clip the priority weights less than 1 to 1, the results will be much better (608.7 v.s. 672.9). However, clipping gives a biased estimation to the weights and hinders the performance of ODPR-A. A more straightforward and effective solution is to apply data prioritization to both improvement and constraint terms.

For applying data prioritization to policy evaluation, we empirically found that usually it severely degrades the performance except for few cases. We hypothesize that data prioritization changes the state-action distribution of the dataset and intensifies the degree of off-policy between the curent policy and the dataset. Although it does not harm policy learning, it might potentially cause instability in policy evaluation when combined with bootstrap and function approximators (Sutton & Barto, 2018; Van Hasselt et al., 2018; Tsitsiklis & Van Roy, 1996). This also explains why ODPR-A is severely impaired by data prioritization for policy evaluation, whereas ODPR-R is not. The underlying cause can be that ODPR-A evaluates the policy on more off-policy data obtained by multiple iterations. Similar effect of decoupled resampling on Antmaze are observed in Appendix D.4.

## 4.4 THE BENEFIT OF FINE-GRAINED PRIORITIES

Existing resampling works such as AW/RW (Hong et al., 2023) and percentage sampling (Chen et al., 2021) assigned the same weights to *all transitions in a trajectory* according to the trajectory return. Instead, ODPR-A resamples each transition according to its advantage, which we term *fine-grained priorities*. Here we will provide more explanation and empirical evidence to illustrate the benefit of fine-grained priorities.

**Applicability to Datasets Lacking Trajectory.** Even when trajectory is unavailable, ODPR-A can be employed due to its fine-grained property. To illustrate the benefit of ODPR-A in such a scenario, we constructed transition-based datasets by randomly sampling 50% of transitions

Table 6: ODPR-A on 50% Transitions.

| 50% Dataset | TD3+BC | | IQL | |
|-------------|--------|------|-----|------|
| 10 seeds    | V      | A    | V   | A    |
| halfcheetah-m | 48.5 | **49.9** | 44.1 | **47.3** |
| hopper-m    | 60.4   | **72.9** | 57.0 | **68.8** |
| walker2d-m  | 66.0   | **82.5** | 67.7 | **81.7** |
| halfcheetah-mr | 43.2 | **45.2** | 35.5 | **39.2** |
| hopper-mr   | 76.4   | 74.9    | 78.5 | **91.2** |
| walker2d-mr | 53.3   | **89.6** | 55.6 | **75.2** |
| halfcheetah-me | **89.1** | 77.7 | 90.8 | 89.9 |
| hopper-me   | 100.0  | **108.4** | 106.9 | **108.9** |
| walker2d-me | 110.3  | **111.6** | 109.4 | **112.2** |
| total       | 647.2  | **712.7** | 645.5 | **714.4** |
| SD (total)  | 25.2   | 13.1    | 23.9 | 14.7 |

from the D4RL Mujoco datasets. Table 6 demonstrates that ODPR-A improves the total score of TD3+BC from 647.2 to 712.7, enhancing performance in 7 out of 9 tasks. Similar enhancements were observed in the IQL case. This evidence reveals that ODPR-A can significantly boost popular offline RL algorithms even without access to trajectory information.

---

[2]IQL and OnestepRL utilize weighted regression, coupling policy constraint and improvement together.

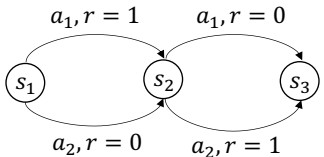

**Concatenating Trajectories to Create Better Behaviors.** The fine-grained priorities also enable ODPR-A to combine favorable segments of suboptimal trajectories, thereby forming improved behavior. This concept is illustrated by Figure 3, where we consider two suboptimal trajectories in the dataset: $\{s_1, a_1, 1, s_2, a_1, 0, s_3\}, \{s_1, a_2, 0, s_2, a_2, 1, s_3\}$. Since two trajectories yield equal returns, AW/RW would not affect the original dataset. In contrast, ODPR-A calculates normalized advantage as priority weights, assigning near-zero weight to $\{s_1, a_2, 0, s2\}, \{s_2, a_1, 0, s_3\}$, and consequently deriving an optimal dataset $\{s_1, a_1, 1, s2\}, \{s_2, a_2, 1, s_3\}$. Moreover, we substantiate the enhanced performance yielded by fine-grained priorities by comparing ODPR-A with AW, employing TD3+BC and IQL in the D4RL Mujoco experiments. As outlined in Table 7, ODPR-A outperforms AW in 8 out of 9 tasks in both TD3+BC and IQL, highlighting its effectiveness in achieving superior performance.

Figure 3: The toy example for trajectory concatenation.

### 4.5 COMPARISON WITH ONLINE SAMPLING METHODS.

In this section, we compare ODPR with representative PER (Schaul et al., 2016), which aims to accelerate value function fitting by dynamically employing the **absolute** TD-error as the priority. Consider a sample with large **negative** TD errors (*i.e.*, advantage), PER gives high priorities to them while ODPR discourages them. Specifically, PER thinks the sample contains more information for value fitting, while ODPR thinks the sample's action is not good behavior. In Figure 4, every curve is an average of 9 mujoco locomotion tasks. The results show that PER slightly harms TD3+BC in offline Mujoco domains. In contrast, ODPR greatly enhances the performance.

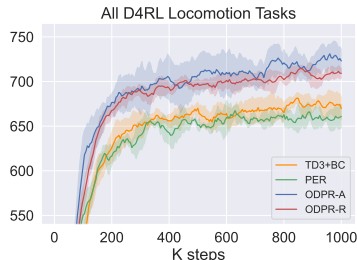

Figure 4: Compare ODPR and PER on offline Mujoco.

### 4.6 COMPARISON WITH OFFLINE SAMPLING METHODS.

In Section 4.4, we demonstrate that fine-grained priorities enables ODPR-A to maintain applicability without trajectory and concatenate suboptimal segments, two features not shared by return-based methods such as AW/RW and percentage sampling. In this section, we further elucidate the distinctions between ODPR and these methods.

A vital differentiation between ODPR and AW/RW lies in decoupled resampling. Unlike AW/RW, ODPR-R does not prioritize policy evaluation, opting instead to employ two separate samplers. As we verify in Section 4.3, prioritizing policy evaluation can intensify the degree of off-policy and lead to potential performance degradation. The AW column of Table 3b illustrates this point, where we compared ODPR-R/ODPR-A and AW using the default hyperparameters of IQL for a fair comparison. In Antmaze, while AW outperformed ODPR slightly in two simpler tasks, it significantly lagged in four more challenging tasks, resulting in a substantial deficit in total performance. This contrast highlights the value of decoupled resampling, an approach unique to ODPR.

Percentage sampling runs algorithms such as TD3+BC and IQL on only top X% data ordered by trajectory returns. In Figure 5, we tested values of 1%, 10%, 25%, 50%, and 75% and found that 50% is nearly the optimal value for both TD3+BC and IQL. However, 50% percentage sampling still underperforms ODPR-A/R (695.2 v.s. 727.8/726.7 in IQL; 687.6 v.s. 734.1/707.3 in TD3+BC), further highlighting the effectiveness of ODPR.

## 5 CONCLUSION AND LIMITATION

This paper proposes a plug-and-play component ODPR for offline RL algorithms by prioritizing data according to our proposed action-quality priority function and decoupled sampling. Furthermore, we theoretically show that a better policy constraint is likely induced when the proposed ODPR is used. We develop two practical implementations, ODPR-A and ODPR-R, to compute priority. Extensive experiments demonstrate that both ODPR-A and ODPR-R can effectively boost the performance of popular RL algorithms. The iterative computation of ODPR-A adds an extra computational burden. In this paper, ODPR-A mitigates this issue by sharing weights across different algorithms. Exploring more efficient methods for obtaining priority weights remains an avenue for future work.

## 6 ETHICS STATEMENT

In the paper, we develop a data prioritization component that aims to boost the performance of offline RL algorithms. From this perspective, any negative societal impact that our method may cause is similar to that of general RL algorithms. We advocate that RL-based robotics systems, game AI, and other applications should follow fair and safe principles.

## 7 REPRODUCIBILITY

To ensure the reproducibility of our results, we have included the source code within the supplementary materials and have provided a comprehensive description of the experimental setup in Appendix C. Upon publication, the code and priority weights will be made publicly available on the Internet. We have reported the average results from multiple runs, the majority of which exhibit a relatively small standard deviation. We assure that all the results presented in this paper can be reproduced effortlessly by executing our provided code.

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

APPENDIX

## A    RELATED WORKS

**Offline RL with Behavior Regularization.** To alleviate the distributional shift problem, a general framework employed by prior offline RL research is to constrain the learned policy to stay close to the behavior policy. Many works (Jaques et al., 2019; Wu et al., 2019) opt for KL-divergence as policy constraint. Exponentially advantage-weighted regression (AWR), an analytic solution of the constrained policy search problem with KL-divergence, is adopted by AWR (Peng et al., 2019), CRR (Wang et al., 2020b) and AWAC (Nair et al., 2020). IQL (Kostrikov et al., 2021b) follows AWR for policy improvement from the expectile value function that enables multi-step learning. BEAR (Kumar et al., 2019) utilizes maximum mean discrepancy (MMD) to approximately constrain the learned policy in the support of the dataset, while Wu *et al.* (Wu et al., 2019) find MMD has no gain over KL divergence. Other variants of policy regularization include the use of Wasserstein distance (Wu et al., 2019) and BC (Fujimoto & Gu, 2021; Wang et al., 2023; Chen et al., 2023). An alternative approach to regularize behavior involves modifying the Q-function with conservative estimates (Kumar et al., 2020b; Buckman et al., 2021; Yu et al., 2021; An et al., 2021; Nikulin et al., 2022; Ghasemipour et al., 2022).

**Prioritized Sampling.** Many resampling methods, i.e., prioritization methods, have been proposed for RL in an online setting, including PER (Schaul et al., 2016), DisCor (Kumar et al., 2020a), LFIW (Sinha et al., 2022), PSER (Brittain et al., 2019), ERE (Wang et al., 2020a), and ReMER(Liu et al., 2021b). These methods mainly aim to expedite temporal difference learning. SIL (Oh et al., 2018) only learns from data with a discounted return higher than current value estimate. In offline RL, schemes based on imitation learning (IL) aim to learn from demonstration, naturally prioritizing data with high return. These approaches include data selection (Chen et al., 2020; Liu et al., 2021a) and weighted imitation learning (Wang et al., 2018). BAIL (Chen et al., 2020) estimates the optimal return, based on which good state-action pairs are selected to imitate. For RL-based learning from offline data, CQL (ReDS) (Singh et al., 2022) is specifically designed for CQL to reweight the data distribution. It works via a modified CQL constraint where the values of bad behaviors are being penalized.This approach is non-trivial to transfer to algorithms like TD3+BC or IQL, since they lack existing components to penalize certain actions. In contrast, ODPR serves as a plug-and-play solution, designed to enhance a broad range of offline RL algorithms. ODPR might be the preferable choice for tasks where other SOTA policy constraint algorithms outperform CQL, such as in the Antmaze or computation-limited environments. Hong et al. (2023) and Yue et al. (2022) proposed to reweight the entire trajectories according to their returns. Although sharing some conceptual similarities, our method offers a more fine-grained approach by resampling transitions rather than entire trajectories. Another distinction lies in our use of two samplers for promoting performance and stability, a uniform sampler for policy evaluation, and a prioritized sampler for policy improvement and policy constraint.

---

**Algorithm 2** Advantage-based Offline Decoupled Prioritized Resampling

1: **Require:** Dataset $\mathcal{D} = \{(\boldsymbol{s}, \boldsymbol{a}, \boldsymbol{s}', r)_i\}_{i=1}^N$, *i.e.*, behavior policy $\beta^{(0)}$, the number of iterations $K$, standard deviation $\sigma$, and a policy-constrained algorithm $\mathcal{I}$
2: **Stage1:** Initialize transition priorities $\omega_{i=1}^N = 1$
3: **for** step $k$ in $\{1, \ldots, K\}$ **do**
4:     Evaluate advantage $A^{(k-1)}$ of behavior policy $\beta^{(k-1)}$ by sampling transition $i$ with the probability $\omega_i$.
5:     Calculate $\omega(A^{(k-1)}(\boldsymbol{s}_i, \boldsymbol{a}_i))$ by Equation (6).
6:     $\omega_i := \omega_i * \omega(A^{(k-1)}(\boldsymbol{s}_i, \boldsymbol{a}_i))$
7: **end for**
8: Scale the standard deviation of $\omega_i$ to $\sigma$.
9: **Stage2:** Train algorithm $\mathcal{I}$ on dataset $\mathcal{D}$. Sample transition $i$ with the priority $\omega_i$ for policy constraint and improvement. Uniform sample for policy evaluation.

---

# B  PROOF AND DERIVATION

## B.1  PROOF OF PERFORMANCE DIFFERENCE LEMMA

Assume that $\pi$ and $\beta$ are two arbitrary policies. $p(s_0)$ denotes the initial state distribution independent of policy. According to previous works (Kakade & Langford, 2002), we have:

$$J(\pi) - J(\beta) \tag{10}$$

$$= \mathbb{E}_{\tau \sim p_\pi(\tau)} \left[ \sum_{t=0}^{\infty} \gamma^t r(s_t, a_t) \right] - \mathbb{E}_{s_0 \sim p(s_0)} \left[ V^\beta(s_0) \right] \tag{11}$$

$$= \mathbb{E}_{\tau \sim p_\pi(\tau)} \left[ \sum_{t=0}^{\infty} \gamma^t r(s_t, a_t) - V^\beta(s_0) \right] \tag{12}$$

$$= \mathbb{E}_{\tau \sim p_\pi(\tau)} \left[ \sum_{t=0}^{\infty} \gamma^t \left( r(s_t, a_t) + \gamma V^\beta(s_{t+1}) - V^\beta(s_t) \right) \right] \tag{13}$$

$$= \mathbb{E}_{\tau \sim p_\pi(\tau)} \left[ \sum_{t=0}^{\infty} \gamma^t A^\beta(s_t, a_t) \right] \tag{14}$$

$$= \sum_{t=0}^{\infty} \int_s p(s_t = s|\pi) \int_a \pi(a|s) \gamma^t A^\beta(s, a) \, da \, ds \tag{15}$$

$$= \int_s \sum_{t=0}^{\infty} \gamma^t p(s_t = s|\pi) \int_a \pi(a|s) A^\beta(s, a) \, da \, ds \tag{16}$$

$$= \int_s d_\pi(s) \int_a \pi(a|s) A^\beta(s, a) \, da \, ds, \tag{17}$$

where $d_\pi(s) = \sum_{t=0}^{\infty} \gamma^t p(s_t = s|\pi)$ represents the unnormalized discounted state marginal distribution induced by the policy $\pi$, and $p(s_t = s|\pi)$ is the probability of the state $s_t$ being $s$ when following policy $\pi$.

## B.2  SOLUTION DERIVATION OF CONSTRAINED POLICY SEARCH PROBLEM

Following AWR, we simplify the constrained policy search problem, relaxing the hard KL constraint by converting it into a soft constraint with coefficient $\lambda$. So we can formulate the Lagrangian function:

$$\mathcal{L}(\pi, \beta, \alpha) = \int_s d_\beta(s) \int_a \pi(a|s) A^\beta(s, a) \, da \, ds + \lambda \left( \epsilon - \int_s d_\beta(s) \mathrm{D}_{\mathrm{KL}} \left( \pi(\cdot|s) || \beta(\cdot|s) \right) ds \right)$$
$$+ \int_s \alpha_s \left( 1 - \int_a \pi(a|s) da \right) ds, \tag{18}$$

By solving the Lagrangian function, setting $\frac{\partial \mathcal{L}}{\partial \pi} = 0$, the optimal policy is given by

$$\pi^*(a|s) = \frac{1}{Z(s)} \beta(a|s) \exp \left( \frac{1}{\lambda} A^\beta(s, a) \right), \tag{19}$$

$Z$ is given by

$$Z(s) = \int_a \beta(a|s) \exp \left( \frac{1}{\lambda} A^\beta(s, a) \right) da. \tag{20}$$

## B.3  PROOF OF THEOREM 3.1

**Behavior Policy Improvement Guarantee.**  Following Performance Difference Lemma (Equation (2)), we have

$$J(\beta') - J(\beta) = \int_s d_{\beta'}(s) \int_a \beta'(a|s) A^\beta(s, a) \, da \, ds. \tag{21}$$

For simplicity, we use $A^\beta$ instead of $A^\beta(\boldsymbol{s}, \boldsymbol{a})$. The inner integral is:

$$\int_{\boldsymbol{a}} \beta'(\boldsymbol{a}|\boldsymbol{s}) A^\beta \, d\boldsymbol{a} \tag{22}$$

$$= \int_{\boldsymbol{a}} \frac{\omega(A^\beta)\beta(\boldsymbol{a}|\boldsymbol{s})}{\int_{\boldsymbol{a}} \omega(A^\beta)\beta(\boldsymbol{a}|\boldsymbol{s})d\boldsymbol{a}} A^\beta \, d\boldsymbol{a} \tag{23}$$

$$= \frac{\int_{\boldsymbol{a}} \left(\omega(A^\beta) - \omega(0)\right)\beta(\boldsymbol{a}|\boldsymbol{s})A^\beta \, d\boldsymbol{a}}{\int_{\boldsymbol{a}} \omega(A^\beta)\beta(\boldsymbol{a}|\boldsymbol{s})d\boldsymbol{a}}. \tag{24}$$

The derivation from Equation (23) to Equation (24) utilizes the property of advantage $\int_{\boldsymbol{a}} A^\beta \beta(\boldsymbol{a}|\boldsymbol{s})d\boldsymbol{a}$ = 0 and $\omega(0)$ is a constant with respect to action. The sign of the integrand in Equation (24) depends on $\left(\omega(A^\beta) - \omega(0)\right) A^\beta$. Since $\omega(A^\beta)$ is monotonic increasing with respect to $A^\beta$, $A^\beta$ and $\omega(A^\beta) - \omega(0)$ have an identical sign. The integrand is always non-negative, which implies that $J(\beta') - J(\beta) \geq 0$ always holds. If there exists a state $\boldsymbol{s}$, under which not all actions in action support $\{\boldsymbol{a}|\beta(\boldsymbol{a}|\boldsymbol{s}) > 0, \boldsymbol{a} \in \mathcal{A}\}$ have zero advantage, the inequation strictly holds. By the definition of advantage, all actions have zero advantage if and only if all actions have the same $Q$-value. To summarize, Theorem 3.1 suggests that policy improvement is ensured if the current policy is weighted according to its normalized advantage. This concept echoes the core principle of policy gradient methods that optimize the likelihood of actions in proportion to the magnitude of their advantage.

### B.4 PROOF OF LEARNED POLICY IMPROVEMENT UNDER ASSUMPTIONS

In this section, we prove that the learned policy $\pi'^*$ is better than $\pi^*$ when assuming $d_{\beta'}(\boldsymbol{s}) = d_\beta(\boldsymbol{s})$. Let $\omega(A^\beta(\boldsymbol{s}, \boldsymbol{a})) = A^\beta(\boldsymbol{s}, \boldsymbol{a}) - \min_{(\boldsymbol{s}, \boldsymbol{a})} A^\beta(\boldsymbol{s}, \boldsymbol{a})$, we need to prove

$$\hat{\eta}(\pi'^*, \beta) - \hat{\eta}(\pi^*, \beta) \geq 0, \tag{25}$$

where the analytic solution of $\pi^*$ is given in Appendix B.2. Then we have:

$$\hat{\eta}(\pi', \beta)$$
$$= J(\pi') - J(\beta)$$
$$= \int_{\boldsymbol{s}} d_{\pi'}(\boldsymbol{s}) \int_{\boldsymbol{a}} \pi'(\boldsymbol{a}|\boldsymbol{s}) A^\beta(\boldsymbol{s}, \boldsymbol{a}) \, d\boldsymbol{a} \, d\boldsymbol{s}$$
$$\approx \int_{\boldsymbol{s}} d_{\beta'}(\boldsymbol{s}) \int_{\boldsymbol{a}} \pi'(\boldsymbol{a}|\boldsymbol{s}) A^\beta(\boldsymbol{s}, \boldsymbol{a}) \, d\boldsymbol{a} \, d\boldsymbol{s},$$

Since $\pi'$ is constrained to be close to the new behavior policy $\beta'$, the last step approximation above holds (Schulman et al., 2015; Peng et al., 2019) Therefore, following the similar process in Appendix B.2, the analytic solution of $\pi'^*$ is

$$\pi'^*(\boldsymbol{a}|\boldsymbol{s}) = \frac{1}{Z'(\boldsymbol{s})} \beta'(\boldsymbol{a}|\boldsymbol{s}) \exp\left(\frac{1}{\lambda} A^\beta\right), \tag{26}$$

Note that in Equation (26), the first term $\beta'(\boldsymbol{a}|\boldsymbol{s})$ comes from constraining the learned policy to be near around $\beta'$, while the second term $\exp\left(\frac{1}{\lambda} A^\beta\right)$ comes from the performance baseline $\beta$.

$Z'$ is given by

$$Z'(\boldsymbol{s}) = \int_{\boldsymbol{a}} \beta'(\boldsymbol{a}|\boldsymbol{s}) \exp\left(\frac{1}{\lambda} A^\beta\right) d\boldsymbol{a} = \frac{\int_{\boldsymbol{a}} \omega(\boldsymbol{a}, \boldsymbol{s})\beta(\boldsymbol{a}|\boldsymbol{s})\exp\left(\frac{1}{\lambda} A^\beta\right) d\boldsymbol{a}}{\int_{\boldsymbol{a}} \omega(\boldsymbol{a}, \boldsymbol{s})\beta(\boldsymbol{a}|\boldsymbol{s})d\boldsymbol{a}}. \tag{27}$$

Combining the analytic solution of $\pi^*$ and $\pi'^*$, and the definition of $\hat{\eta}(\pi'^*, \beta)$ and $\hat{\eta}(\pi^*, \beta)$, the goal is equivalent to proving

$$\int_{\boldsymbol{s}} d_{\beta'}(\boldsymbol{s}) \int_{\boldsymbol{a}} \frac{1}{Z'(\boldsymbol{s})} \beta'(\boldsymbol{a}|\boldsymbol{s})\exp\left(\frac{1}{\lambda} A^\beta\right) A^\beta \, d\boldsymbol{a} \, d\boldsymbol{s}$$
$$- \int_{\boldsymbol{s}} d_\beta(\boldsymbol{s}) \int_{\boldsymbol{a}} \frac{1}{Z(\boldsymbol{s})} \beta(\boldsymbol{a}|\boldsymbol{s})\exp\left(\frac{1}{\lambda} A^\beta\right) A^\beta \, d\boldsymbol{a} \, d\boldsymbol{s} > 0. \tag{28}$$

According to the assumption, we have $d_{\beta'}(s) = d_\beta(s)$. Therefore, we can ignore the outer integration with respect to $s$ in Equation (28) for a moment. We just consider

$$\int_a \frac{1}{Z'(s)} \beta'(a|s) \exp\left(\frac{1}{\lambda} A^\beta\right) A^\beta \, da - \int_a \frac{1}{Z(s)} \beta(a|s) \exp\left(\frac{1}{\lambda} A^\beta\right) A^\beta \, da \tag{29}$$

$$= \int_a \exp\left(\frac{1}{\lambda} A^\beta\right) A^\beta \left(\frac{\beta'(a|s)}{Z'(s)} - \frac{\beta(a|s)}{Z(s)}\right) \, da \tag{30}$$

$$= \int_a \exp\left(\frac{1}{\lambda} A^\beta\right) A^\beta \beta(a|s) \left(\frac{\omega(s,a)}{\int_a \omega(s,a)\beta(a|s)\exp\left(\frac{1}{\lambda}A^\beta\right)da} - \frac{1}{\int_a \beta(a|s)\exp\left(\frac{1}{\lambda}A^\beta\right)da}\right) \, da \tag{31}$$

$$= \frac{\mathcal{L}(\beta)}{\left(\int_a \omega(s,a)\beta(a|s)\exp\left(\frac{1}{\lambda}A^\beta\right)da\right)\left(\int_a \beta(a|s)\exp\left(\frac{1}{\lambda}A^\beta\right)da\right)} \tag{32}$$

$$= C(s)\mathcal{L}(\beta). \tag{33}$$

where $C(s)$ is the reciprocal of the denominator in Equation (32), obviously a positive constant with respect to $a$, and $\mathcal{L}(\beta)$ is:

$$\mathcal{L}(\beta) = \left(\int_a \beta(a|s)\exp\left(\frac{1}{\lambda}A^\beta\right)da\right)\left(\int_a \beta(a|s)A^\beta\omega(s,a)\exp\left(\frac{1}{\lambda}A^\beta\right)da\right)$$
$$- \left(\int_a \beta(a|s)\omega(s,a)\exp\left(\frac{1}{\lambda}A^\beta\right)da\right)\left(\int_a \beta(a|s)A^\beta\exp\left(\frac{1}{\lambda}A^\beta\right)da\right). \tag{34}$$

When the priority function is linear, we have

$$\mathcal{L}(\beta) = \left(\int_a \beta(a|s)\exp\left(\frac{1}{\lambda}A^\beta\right)da\right)\left(\int_a \beta(a|s)A^\beta(A^\beta - \min_{(s,a)} A^\beta)\exp\left(\frac{1}{\lambda}A^\beta\right)da\right)$$
$$- \left(\int_a \beta(a|s)(A^\beta - \min_{(s,a)} A^\beta)\exp\left(\frac{1}{\lambda}A^\beta\right)da\right)\left(\int_a \beta(a|s)A^\beta\exp\left(\frac{1}{\lambda}A^\beta\right)da\right). \tag{35}$$

Simplify the terms:

$$\mathcal{L}(\beta) = \left(\int_a \beta(a|s)\exp\left(\frac{1}{\lambda}A^\beta\right)da\right)\left(\int_a \beta(a|s)A^\beta A^\beta\exp\left(\frac{1}{\lambda}A^\beta\right)da\right) - \left(\int_a \beta(a|s)A^\beta\exp\left(\frac{1}{\lambda}A^\beta\right)da\right)^2. \tag{36}$$

By Cauchy–Schwarz inequality, we have

$$\left(\int_a \beta(a|s)\exp\left(\frac{1}{\lambda}A^\beta\right)da\right)\left(\int_a \beta(a|s)A^\beta A^\beta\exp\left(\frac{1}{\lambda}A^\beta\right)da\right)$$
$$\geq \left\{\int_a \sqrt{\beta(a|s)\exp\left(\frac{1}{\lambda}A^\beta\right)}\sqrt{\beta(a|s)A^\beta A^\beta\exp\left(\frac{1}{\lambda}A^\beta\right)}da\right\}^2 \tag{37}$$
$$= \left(\int_a \beta(a|s)A^\beta\exp\left(\frac{1}{\lambda}A^\beta\right)da\right)^2.$$

Thus we have $\mathcal{L}(\beta) \geq 0$. Equality is achieved if and only if $A^\beta(s,a)A^\beta(s,a)$ is a constant for any action in action support $\mathcal{B} = \{a|\beta(a|s) > 0, a \in \mathcal{A}\}$, i.e., $A^\beta(s,a) = 0, \quad \forall a \in \mathcal{B}$. So LHS of Equation (25) is equivalent to

$$\int_s d_\beta(s) \int_a C(s)\mathcal{L}(\beta) \, da \, ds, \tag{38}$$

which is obviously non-negative. That is to say the target of Equation (25) holds. The inequality strictly holds when there exists a state $s$, under which not all actions in action support $\mathcal{B}$ have the same $Q$-value.

## C  EXPERIMENT SETUP

### C.1  TOY BANDIT

The offline dataset is collected by four 2D Gaussian distribution policies with means $\boldsymbol{\mu} \in \{(0.6, 0.6), (0.6, -0.6), (-0.6, 0.6), (-0.6, -0.6)\}$, standard deviations $\boldsymbol{\sigma} = (0.05, 0.05)$, and correlation coefficient $\rho = 0$. The reward of each action is sampled from a Gaussian distribution, whose mean is determined by its action center and the standard deviation is 0.5. Each policy contributes 2500 samples to the dataset, imitating real scenarios where various policies collect data.

### C.2  ODPR

**ODPR-A experiment setup.**  In Equation (7), we adopt double value network to fit the value function  (Hasselt, 2010; Fujimoto et al., 2018). Specifically, we employ a one-step bootstrap for value function fitting. For each iteration in value function fitting, we utilize 0.5M gradient steps. Pure value fitting comsumes less computation time than policy iteration. While this lengthy number of gradient steps ensures convergence, a lesser number may suffice in practice. For Mujoco tasks, BC and TD3+BC prioritize data using priority weights from the 4th and 5th iterations, respectively, achieving optimal results (refer to Table 4). For other algorithms, *i.e.*, CQL, IQL, and OnestepRL, we do not perform hyperparameter search and, following BC, simply set the number of iterations to 4.

**Scale Deviation.**  ODPR-A utilizes Equation (6) to calculate weights. However, most of the weights are close to 1, weakening the effect of data prioritization. Therefore, we scale the standard deviation of weights to the hyperparameter $\sigma$ by the following equation:

$$\omega(A^{\beta}(\boldsymbol{s}, \boldsymbol{a})) = \min(\frac{\omega(A^{\beta}(\boldsymbol{s}, \boldsymbol{a})) - 1}{\sigma_0} * \sigma + 1, 0.1),$$

where $\sigma_0$ is the standard deviation of the unscaled weights. After scaling, the mean and standard deviation of the weights are approximately equal to 1 and sigma, respectively. Across all Mujoco tasks, we set $\sigma$ to 2.0. For Antmaze tasks, we use $\sigma = 5.0$, while for Kitchen and Pen tasks, we set $\sigma = 0.5$.

**ODPR-R experiment setup.**  The base priority $p_{\text{base}}$ in Equation (9) is set to zero across all tasks, with the exception of Antmaze where it is assigned a value of 0.2. This exception is made because the trajectory return in Antmaze can either be zero or one. If $p_{\text{base}}$ were set to zero, all trajectories with a return of zero would be disregarded.

**Resampling Implementation.**  PER (Schaul et al., 2016) employs a dynamic update of priorities during training and implements a sophisticated "sum-tree" structure to optimize efficiency. Conversely, ODPR first computes priorities from the offline dataset in an initial stage, and subsequently employs static priorities throughout training. This static priority sampling is straightforwardly implemented using "np.random.choice". While the utilization of a sum tree allows for sampling from a list of probabilities in $O(logn)$ time, an aspect independent of online versus offline scenarios, the use of "np.random.choice" incurs a linear $O(n)$ cost for sampling from the same list of probabilities. However, it's important to note that the single operation of np.random.choice" is time-efficient. Additionally, since ODPR's priorities are static, we can pre-sample the entire index list, thus accelerating the sampling process due to the parallelized implementation of np.random.choice". In the context of training on a D4RL dataset (comprising 1M data points with priority, 1M gradient steps, and a batch size of 256), it only takes approximately 53.8 seconds to execute all "np.random.choice" operations. This adds a negligible time cost to the overall training process. Furthermore, the implementation of ODPR demands merely 10 lines of code changes, which contrasts with the more complex implementation of a sum tree. Therefore, from a practical standpoint, implementing "np.random.choice" is fairly straightforward and only marginally increases the actual runtime. Lastly, for larger dataset sizes (such as 10M or 100M) and a larger number of gradient steps, the additional time cost can be further mitigated by parallelizing the index generation process with the agent training process.

**Computational Cost.**  It's important to note that the priority weights generated by ODPR-A can be reused across different algorithms. Indeed, all the algorithms utilized in this study (including

TD3+BC, IQL, etc.) utilize the same set of weights. Furthermore, training the value network to estimate priority in ODPR-A (when K=5) does not necessitate five times the computational resources compared to standard TD3+BC. Contrasting ODPR-A to TD3+BC, ODPR-A requires no actor update and does not need to query the actor during policy evaluation, making it less time-intensive per step. In our tests conducted on an NVIDIA 3090, 1M gradient steps took 69 minutes for the official TD3+BC, whereas our JAX implementation of ODPR-A took only 7 minutes for 0.5M steps. Thus, even with the number of iterations set to $K = 5$, ODPR-A priority estimation takes less time (35 minutes) than TD3+BC (69 minutes). In summary, ODPR-A does not consume excessive time.

## C.3 OFFLINE RL ALGORITHMS

For a fair comparison with the baselines, we implement ODPR on top of the official implementation of OnestepRL, TD3+BC, and IQL; for CQL, we use a reliable third-party implementation[3], which, unfortunately, causes a slight discrepancy with PyTorch version results reported in the CQL paper. We run every algorithm for 1M gradient steps and evaluate it every 5,000 steps, except for antmaze-v0 environments, in which we evaluate every 100,000 steps. For antmaze-v0, each evaluation contains 100 trajectories; for others, 10 trajectories are used following Kostrikov et al., 2021b. For OnestepRL, we choose exponentially weighted regression as the policy improvement operator. The original paper (Brandfonbrener et al., 2021) does not specify the value of temperature $\tau$. Therefore, we search on a small set $\{0.1, 0.3, 1.0, 3.0, 10.0, 30.0\}$ and use $\tau = 1$ because it can reproduce the reported results.

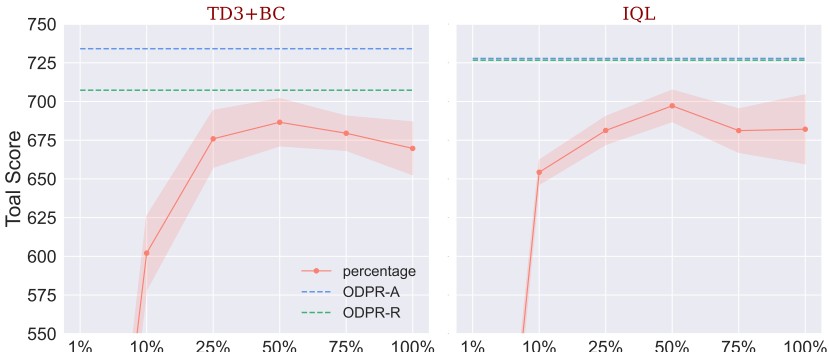

Figure 5: Compare ODPR and percentage sampling on mujoco locomotion based on TD3+BC and IQL. 50% is the optimal value for percentage sampling, still far behind ODPR.

Table 7: Compare ODPR-A with AW. AW resamples all transitions in a trajectory with the same probability without fine-grained priorities. We maintained identical hyperparameters for both ODPR-A and AW to ensure a fair comparison. ODPR-A achieves the best score in 8 out of 9 tasks.

|  | TD3+BC | | | IQL | | |
|---|---|---|---|---|---|---|
|  | vanilla | ODPR-A | AW | vanilla | ODPR-A | AW |
| halfcheetah-m | 48.2 | **50.0** | 48.6 | **47.6** | **47.5** | 45.1 |
| hopper-m | 58.8 | **74.1** | 61.1 | 64.3 | **73.5** | 62.7 |
| walker2d-m | 84.3 | **84.9** | 80 | 79.9 | **83.1** | 76.1 |
| halfcheetah-mr | 44.6 | **45.9** | 45.1 | 43.4 | **44.1** | 42.2 |
| hopper-mr | 58.1 | **88.7** | 87.4 | 89.1 | **103** | 93.3 |
| walker2d-mr | 73.6 | **88.2** | 80.3 | 69.6 | **81** | 62.6 |
| halfcheetah-me | 93.0 | 83.3 | **97.7** | 83.5 | 88.9 | **93.7** |
| hopper-me | 98.8 | **107.3** | 102.9 | 96.1 | **100.3** | 93.3 |
| walker2d-me | 110.3 | **111.7** | 110.2 | 109.2 | **111.4** | 108.8 |
| total | 669.7 | **734.1** | 708.7 | 682.7 | **732.8** | 677.8 |

---

[3] https://github.com/young-geng/JaxCQL

# D ADDITIONAL EXPERIMENTS

## D.1 HOW DOES ODPR PERFORM WHEN THE BEHAVIOR POLICY IS A MIXTURE OF POLICIES

To validate our core proposition, we also constructed a mixed dataset by mixing random and expert datasets. Table 8(a) reflects that the vanilla TD3+BC underperforms within these mixed datasets. In contrast, ODPR-R manages to target beneficial actions, resulting in a performance that stands on par with a pure expert.

## D.2 HOW DOES ODPR PERFORM WHEN THE BEHAVIOR POLICY EXHIBITS SIMILAR AND POOR PERFORMANCE

We ran a comparative study using vanilla TD3+BC and the ODPR-R variation on three D4RL random datasets. As displayed in Table 8(b), ODPR's performance mirrors the vanilla. This observation aligns with our key hypothesis that ODPR's enhancements are primarily sourced from the behavioral diversity inherent in the dataset. In contrast, the random dataset, generated through a poor policy, exhibits limited behavioral variance.

Table 8: Performance of ODPR-R on random and mixed datasets with standard deviation (10 seeds). (a) Left: Random Dataset. (b) Right: Mixed dataset of random and expert.

| Mixed Dataset | TD3+BC | | Random Dataset | TD3+BC | |
| --- | --- | --- | --- | --- | --- |
| | Vanilla | ODPR-R | | Vanilla | ODPR-R |
| halfcheetah | 89.3 | **96.6** | halfcheetah | 9.8 | 10.3 |
| hopper | 102.2 | **108.7** | hopper | 8.4 | 8.3 |
| walker2d | 25.6 | **110.0** | walker2d | 0.9 | 1.0 |
| total | 217.1± 11.8 | **315.3± 4.7** | total | 19.1± 1.8 | 19.6± 1.6 |

## D.3 HOW DOES ODPR PERFORM WHEN VALUE ESTIMATIONS ARE INACCURATE

According to DisCor (Kumar et al., 2020a), value-based techniques can suffer from a lack of corrective feedback in some cases such as a tree-structured MDP. Although value estimation in Equation (7) works well in D4RL environments, we investigate how does ODPR-A performs when value estimations are inaccurate, which bears significance for the application of ODPR-A in scenarios without corrective feedback. In this context, we tested ODPR-A's performance when the value estimations are inaccurate by adding independent Gaussian noise $N(0, sx)$ to priority weights, where $x$ represents the average distance between the mean point and all weights. We set $s$ to $0.1, 1, 2, 5$ and test TD3+BC with ODPR-A with two environments with 10 seeds. These two environments highlight situations where ODPR-A's performance is either comparable to or better than the standard method. Please refer to Table 9. It's worth noting that when noise is as large as the average distance $x$, ODPR-A's performance remains nearly unimpaired. Even when the noise is five times $x$ on hopper-medium-replay, ODPR still outperforms the vanilla. On walker-medium-expert, ODPR demonstrates stability despite varying noise levels. The empirical findings suggest that ODPR-A retains effectiveness even with relatively substantial noise in estimating priority.

Table 9: The performance of ODPR-A with TD3+BC when the priority weights are added with Gaussian noise $N(0, sx)$ to priority weights, where $x$ represents the average distance between the mean point and all weights.

| | Vanilla | ODPR-A ($s$) | | | | |
| --- | --- | --- | --- | --- | --- | --- |
| | | 0 | 0.1 | 1.0 | 2.0 | 5.0 |
| hopper-medium-replay | $58.0 \pm 5.8$ | $88.7 \pm 5.5$ | $88.4 \pm 4.7$ | $88.3 \pm 7.4$ | $74.6 \pm 3.4$ | $73.0 \pm 7.2$ |
| walker2d-medium-expert | $110.2 \pm 0.2$ | $111.7 \pm 0.2$ | $111.5 \pm 0.2$ | $111.3 \pm 0.4$ | $110.7 \pm 0.3$ | $110.5 \pm 0.3$ |

## D.4 MORE EXPERIMENTS ABOUT DECOUPLED RESAMPLING

In Section 4.3, we conduct experiments on locomotion tasks to analyze the effect of prioritizing for different terms and conclude that prioritizing data for both policy improvement and policy constraint is crucial for high performance. Not prioritizing data for policy evaluation is also crucial

for limiting the degree of off-policy and remaining stable. Here, we further analyze the influence of prioritizing data for policy evaluation on other domains and report results in Table 11. We observe a phenomenon analogous to that in Table 3b, wherein prioritizing for policy evaluation significantly impairs performance. The prioritization of data for evaluation markedly compromises the version with decoupled resampling, culminating in near-zero scores in four out of the six tasks. In conclusion, the prioritization of policy evaluation detrimentally affects both resampling methods, whether by return or advantage.

As shown in Section 4.3, for ODPR-A, the best results are obtained by decoupled resampling, where two samplers are employed, one for uniform sampling and one for prioritized sampling. We conduct experiments here to prove that the introduction of two samplers itself does not improve performance, and the improvement comes from prioritized replay. The original TD3+BC with one uniform sampler scored 667.7 points on Mujoco locomotion. Then we use two uniform samplers for the actor (policy constraint and improvement) and critic (policy evaluation), respectively. The result (674.9 points) is quite similar to the vanilla one. Then we use a prioritized sampler

Table 10: Two samplers.

|  | mujoco-v2 total |
|---|---|
| vanilla | 667.7 |
| two samplers | 674.9 |
| ODPR-A | 734.1 |

for the actor and a uniform sampler for the critic, respectively, achieving a high score of 734.1 points. It implies that the improvement comes from prioritized replay rather than two samplers.

Table 11: Effect of decoupled resampling. "-EVAL" denotes data prioritization for only policy constraint and policy improvement terms (*i.e.*, decoupled sampling). "all" denotes data prioritization for all three terms.

|  | vanilla IQL | ODPR-A | | ODPR-R | |
|---|---|---|---|---|---|
|  |  | -EVAL | all | -EVAL | all |
| antmaze-umaze | 88.5 | 85.5 | 77.3 | 87.8 | **89.2** |
| antmaze-umaze-diverse | 63.1 | **70.8** | 67.7 | 66.0 | **79.8** |
| antmaze-medium-play | 70.5 | **76.1** | 0 | 72.0 | 68.4 |
| antmaze-medium-diverse | 58.5 | **71.8** | 0 | **74.2** | 38.4 |
| antmaze-large-play | 44.1 | 40.0 | 0 | **49.6** | 14.6 |
| antmaze-large-diverse | 42.0 | **48.0** | 0 | 43.0 | 37.6 |
| antmaze total | 366.7 | **392.2** | 145.0 | **392.6** | 328.0 |

## D.5 RESULTS WITH STANDARD DEVIATION

Results with standard deviation are reported in Table 12 and Table 13. For Mujoco locomotion tasks, ODPR consistently achieves a performance boost, as evidenced by the non-overlapping deviation intervals. For tasks with larger variance, such as Antmaze, Kitchen, and Adroit, the performance improvement brought by ODPR is statistically significant according to t-test results.

Table 12: Averaged normalized scores of TD3+BC on MuJoCo locomotion v2 tasks over 15 seeds.

| Dataset | TD3+BC | | |
|---|---|---|---|
|  | V | A | R |
| halfcheetah-m | 48.3 ± 0.1 | **50.0 ± 0.1** | 48.6 ± 0.1 |
| hopper-m | 57.3 ± 1.4 | **74.1 ± 2.8** | 59.1 ± 1.2 |
| walker2d-m | 84.9 ± 0.6 | 84.9 ± 0.3 | 84.2 ± 0.3 |
| halfcheetah-mr | 44.5 ± 0.2 | 45.9 ± 0.4 | 44.6 ± 0.4 |
| hopper-mr | 58.0 ± 5.8 | **88.7 ± 5.5** | **77.4 ± 4.2** |
| walker2d-mr | 72.9 ± 8.7 | **88.2 ± 2.0** | 82.7 ± 2.2 |
| halfcheetah-me | 92.4 ± 0.5 | 83.3 ± 3.0 | 93.9 ± 0.7 |
| hopper-me | 99.2 ± 6.3 | **107.3 ± 4.1** | **106.7 ± 3.1** |
| walker2d-me | 110.2 ± 0.2 | 111.7 ± 0.2 | 110.1 ± 0.1 |
| total | 667.7 ± 18.4 | **734.1 ± 10.4** | **707.3 ± 7.9** |

## D.6 THE EFFECT OF HYPERPARAMETER $\sigma$

Intuitively, if the weights of all transitions are close to 1, our methods degrade to the vanilla offline RL algorithm. Only when the standard deviation of the weights of transitions is relatively large, ODPR

Table 13: Averaged normalized scores of IQL on Antmaze, Kitchen, and Adroit tasks over 15 seeds.

|  | vanilla IQL | ODPR-A | ODPR-R |
|---|---|---|---|
| antmaze-umaze | 88.5±3.0 | 85.5±4.4 | 87.8±3.0 |
| antmaze-umaze-diverse | 63.1±6.4 | **70.8±7.8** | 66±7.8 |
| antmaze-medium-play | 70.5±4.6 | **76.1±5.1** | 72±5.4 |
| antmaze-medium-diverse | 58.5±7.2 | **71.8±6.6** | **74.2±9.4** |
| antmaze-large-play | 44.1±4.6 | 40±5.3 | **49.6±4.0** |
| antmaze-large-diverse | 42±4.7 | 48±4.0 | 43±4.9 |
| total | 366.7±18.2 | **392.2±19.1** | **392.6±20.4** |
| kitchen-complete-v0 | 65.9±8.4 | 64.2±6.1 | 62.7±7.8 |
| kitchen-partial-v0 | 51.4±9.7 | **66.5±13.2** | **69.5±6.9** |
| kitchen-mixed-v0 | 50.3±6.8 | 52.1±6.7 | 49.9±3.3 |
| total | 167.6±16.5 | **182.8±15.9** | **182.1±14.4** |
| pen-human-v0 | 73.1±18.0 | 72.9±15.5 | **83±17.2** |
| pen-cloned-v0 | 42.1±21.1 | **61.2±13.6** | **66.6±21.4** |

can take effect. We observed that the standard deviation of original ODPR-A weights in Equation (8) is typically small, ranging from approximately 0.02 to 0.2 across different environments. In contrast, the standard deviation of ODPR-R weights falls within a suitable range of around 0.3 to 1.0. Thus, ODPR-A needs to be scaled, while ODPR-R can work without scaling. We test the effect of $\sigma$ on three environments where ODPR-A gives the clearest improvements. In Table 14, we demonstrate how the performance of ODPR-A is influenced by the hyperparameter $\sigma$, which the standard deviation of weights will be scaled to. We select 2 as the default value for $\sigma$.

Table 14: Effect of $\sigma$ on ODPR-A. The results come from TD3+BC with 15 seeds. "w.o. scale" denotes disabling scaling.

| $\sigma$ | vanilla | w.o. scale | 0.5 | 2.0 | 4.0 |
|---|---|---|---|---|---|
| hopper-mr | 57.9 | 70.1 | 73.8 | **88.7** | **88.9** |
| walker-mr | 73.1 | 81.9 | 84.9 | **88.2** | 86.6 |
| hopper-me | 98.5 | 99.1 | 106.9 | **107.3** | 105.1 |
| total | 229.5 | 251.1 | 265.6 | **284.2** | **280.6** |

## D.7    RESAMPLING V.S. REWEIGHTING

Resampling and reweighting are statistically equivalent with regards to the expected loss function. We provide implementations for both approaches in ODPR. As seen in Table 15 and Table 16, resampling and reweighting yield comparable scores on Mujoco locomotion, as well as Kitchen and Adroit tasks. These results indicate that both reweighting and resampling can successfully implement ODPR. Further, they suggest that the effectiveness of ODPR does not rely on the specific implementation, but rather arises from the prioritization of data itself. The only exception we encountered is observed in the two Pen tasks, where for IQL with ODPR-R, resampling performed well while reweighting was unable to achieve meaningful scores. Notably, in these two tasks, some priority weights of ODPR-R are extremely large due to the presence of exceptionally high returns in the return distributions (see Figure 7). We hypothesize that these exceedingly large weights may alter the learning rate, thereby affecting the gradient descent process.

Table 15: Compare resampling and reweighting implementations for ODPR on Mujoco locomotion.

|  | TD3+BC (ODPR-A) | | TD3+BC (ODPR-R) | | IQL (ODPR-A) | | IQL (ODPR-R) | |
|---|---|---|---|---|---|---|---|---|
|  | resample | reweight | resample | reweight | resample | reweight | resample | reweight |
| mujoco-v2 total | 734.1 | 740.3 | 707.3 | 705.3 | 727.8 | 725.4 | 726.7 | 727.0 |

Table 16: Compare resampling and reweighting implementations of IQL (ODPR-A) on Adroit and Kitchen.

|  | reweighting | resampling |
|---|---|---|
| kitchen-complete-v0 | 68.0 | 64.2 |
| kitchen-partial-v0 | 61.5 | 66.5 |
| kitchen-mixed-v0 | 45.3 | 52.1 |
| kitchen total | 174.8 | 182.8 |
| pen-human-v0 | 73.9 | 72.9 |
| pen-cloned-v0 | 61.6 | 61.2 |

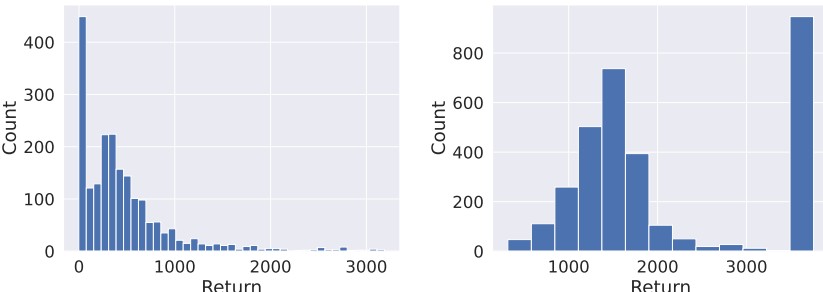

Figure 6: Trajectory Return Distributions of hopper-medium-replay (left) and hopper-medium-expert (right). Medium-replay datasets usually have a long-tailed distribution, and medium-expert often display two peaks. Both are composed of policies with varying quality.

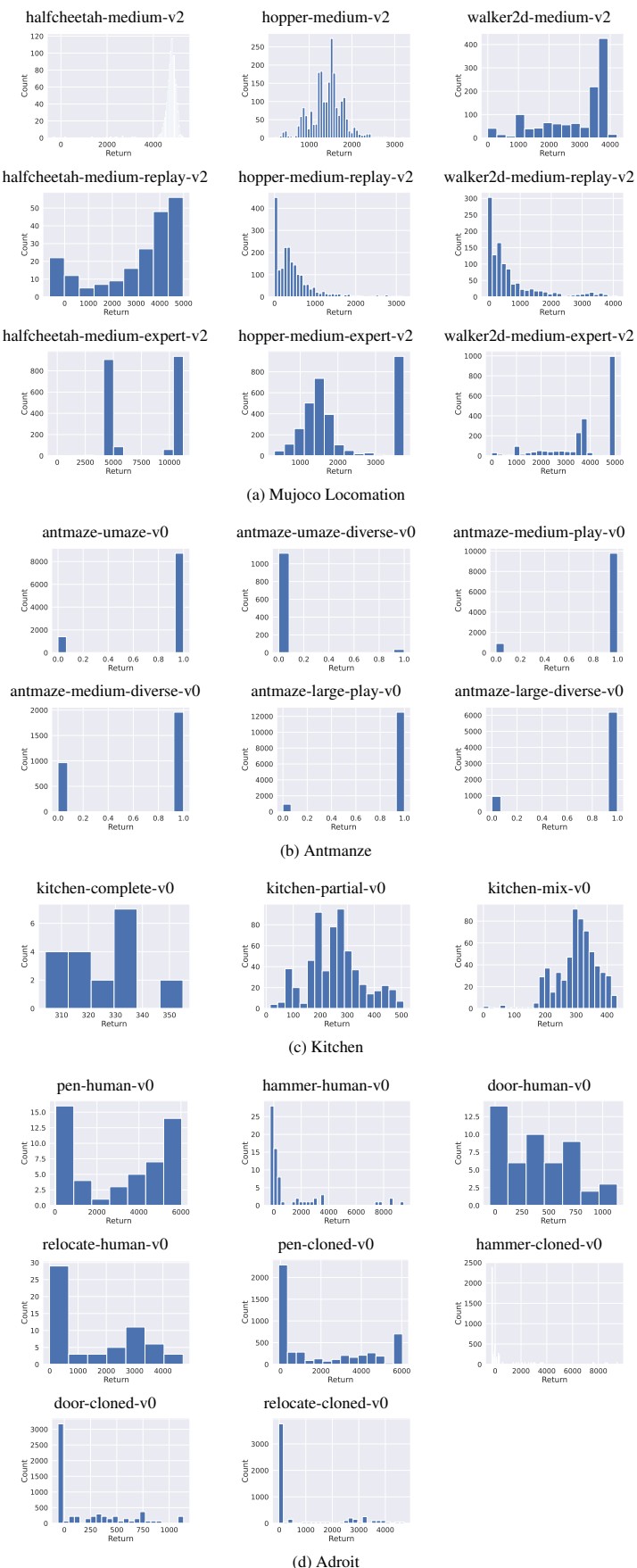

Figure 7: Full Visualization of Trajectory Return Distributions.

