# OpenReview forum: "Decoupled Prioritized Resampling: Advancing Offline RL with Improved Behavior Policy"
_ICLR.cc/2024/Conference — Submitted to ICLR 2024_

### Official Review · Reviewer_sg4V · 2023-10-29

**Soundness:** 1 poor
**Presentation:** 2 fair
**Contribution:** 1 poor
**Rating:** 3
**Confidence:** 4

**Summary:**

This paper proposes a method to resample transitions before offline reinforcement learning. Specifically, they resample transitions in a dataset according to the advantages or the returned rewards. It is assumed that the resampled dataset can be seen as a new dataset that generated by an improved policy, and we can then train a better policy using any offline RL (or BC) algorithm using the resampled dataset.

**Strengths:**

1. The paper is easy to follow. The motivation and the method are described clearly.

**Weaknesses:**

1. The motivation of the paper is inappropriate. As we know, offline RL uses a policy constraint to alleviate the distribution shift problem. However, this paper attempts to first transform the behavior policy $\beta$ to a new strategy $\beta'$ **without any constraint** and assume it is our behavior policy. In this way, the distribution of $\beta'$ and the policy $\pi$ can be arbitrarily different from $\beta$. In other words, $\beta'$ and $\pi$ may query the values of a lot of OOD states, so the distribution shift problem may be serious. Note that the approximation error of the value function on OOD states can be arbitrarily large.
2. The logic behind the idea of resampling is problematic. Except that resampling can make the distribution shift problem serious, the way to do prioritized resampling is problematic. Actually, according to the logic in prioritized resampling, the best prioritized policy should be the pure strategy that always chooses the best action that maximizes $A(s,a)$. But obviously, we can not do that due to the distribution shift problem. Then, the problem is how we can improve $\beta'$ without making it too different from the behavior policy $\beta$. However, this is what offline RL trying to do.
3. The analysis in section 3.2 is unconvincing. **We can not assume $d_{\beta'}(s) = d_{\beta}(s)$**, as the transform from $\beta$ to $\beta'$ is unconstrainted. So the improvement of $\pi$ over $\beta$ is not guaranteed. In this context, the iterative prioritization method proposed in section 3.4 seems meaningless.
4. The experimental results are not convincing. According to my understanding, ODPR eliminates many suboptimal transitions. Although this can improve the performance, it may hurt the generalization. So it would be better if some results about the generalization were provided. Besides, I suspect offline RL with larger model size and more careful hyperparameter tuning may match the performance of ODPR [1].

[1] Tarasov D, Kurenkov V, Nikulin A, et al. Revisiting the Minimalist Approach to Offline Reinforcement Learning[J]. arXiv preprint arXiv:2305.09836, 2023.

**Questions:**

1. Why do we need another resampling/weighting procedure while regularized offline RL (e.g., AWR) actually does the same thing?

---

### Official Review · Reviewer_z7fk · 2023-11-01

**Soundness:** 2 fair
**Presentation:** 2 fair
**Contribution:** 2 fair
**Rating:** 5
**Confidence:** 4

**Summary:**

- For offline RL, previous approaches have used policy constraints to align the learned policy with the behavior policy. The authors argue that these constraints treat well-performing and inferior actions equally through uniform sampling, potentially harming the learned policy.
- In this paper, Offline Decoupled Prioritized Resampling (ODPR) is introduced, which addresses the suboptimal policy constraint issue by designing specialized priority functions and employs decoupled resampling.
- Theoretical analysis shows that these priority functions may improve.
- Two practical implementations are provided: one estimates priorities using a fitted value network (ODPR-A), and the other uses trajectory returns (ODPR-R) for quicker computation.
- Experimental results demonstrate that both ODPR-A and ODPR-R significantly enhance the performance of various baseline methods, including BC, TD3+BC, Onestep RL, CQL, and IQL.

**Strengths:**

- Practical easy-to implement algorithm that can be easily adopted to existing algorithms for improvement
- Various experiments are conducted to prove its efficiency

**Weaknesses:**

- Not much contribution as these resampling methods are already suggested in a similar form in existing literatures (see [1] below)
- Not sure whether this approach is actually very practical, since there are lots of freedom, e.g. weight function choice, hyper parameters like p_base, number of repetitions for odpr-a, etc. and we need to find a best hyper parameters to apply this method, as not well chosen hyper parameters can even hurt the performance (e.g. it may perform poorly if we push iterated odpr-a to the limit). If the online evaluation is expensive, the proposed method will be hard to be applied.

**Questions:**

- Why do we need to do an iterative resampling? Theoretically, importance sampling should have the same impact to resampling, and we can just do importance sampling and the value fitting at the same time, which will allow us to fit weight to converge to the limit of resampling. If it is not desired, we may regularize the learning or early stop to get the same effect of fixed iterations of resampling. What is the benefit of explicit resampling?
- Unlike advantage-based resampling, return based resampling will also alter the state distribution, which may result in undesired effect, e.g. smaller effective state coverage. How are the effective dataset size after each resampling methods?
- I think the paper should compare with [1], because the method seems to be mainly the same, except that they are doing importance sampling and this uses explicit resampling. [1] is already comparing with odpr-r or odpr-a like algorithm and shows that their performance is superior. Does iterative odpr-a perform competitively to [1]?


[1] Beyond Uniform Sampling: Offline Reinforcement Learning with Imbalanced Datasets

---

### Official Review · Reviewer_X3f2 · 2023-11-01

**Soundness:** 3 good
**Presentation:** 3 good
**Contribution:** 2 fair
**Rating:** 5
**Confidence:** 4

**Summary:**

This paper proposes Offline Decoupled Prioritized Resampling (ODPR) as a method to improve offline reinforcement learning (RL) by designing specialized priority functions for the suboptimal policy constraint issue and employing decoupled resampling for training stability.  The authors argue that existing methods for constraint-based offline RL algorithms, which apply constraints equally to all actions through uniform sampling, may negatively affect the learned policy. To alleviate this issue, ODPR introduces a class of priority functions that prioritize highly-rewarding transitions, making them more frequently visited during training. Two strategies are developed for obtaining priority weights based on advantages estimated from a fitted value network (ODPR-A) or trajectory returns (ODPR-R). The performance of ODPR is evaluated on five different algorithms and it is shown to improve the performance for all included baseline methods.

**Strengths:**

1. The paper introduces a novel approach to prioritized replay for offline RL, specifically tailored for addressing the suboptimal policy constraint issue.
2. The theoretical analysis provides a solid foundation for the proposed method and demonstrates the improvement in the behavior policy.
3. The empirical experiments are comprehensive and cover a wide range of domains, showing significant performance improvements across multiple baseline algorithms.

**Weaknesses:**

1. Comparison with AW/RW proposed in [1]: The authors should compare ODPR-R with RW instead of AW. AW uses an advantage estimator of initial states to weight samples, which is different from ODPR-R and RW. To see the benefits of ODPR-R over RW, this comparison is essential. Also, the hyperparameters of ODPR-A is tuned per domain (e.g., MuJoCo, kitchen, antmaze, adroit) but the baselines  AW/RW didn't use the optimized hyperparameters for each domain. This makes the comparsion questionable because the performance gain of ODPR may depend on hyperparameter tuning. The authors should also rerun AW/RW in the same codebase with ODPR instead of taking numbers directly from their codebase.
2. Comparison with AWAC: Assigning higher weights on good transitions is similar to AWAC [2]. Why will the advantage-based reweighting be better than AWAC? There seems to be no comparison in the related work or experiment section. Also, how will OPER behave when the offline behavior policy has similar and poor performance?
3. Lack of baselines: The paper aims to mitigate the issue of imposing regularization equally to each action, including inferior ones. Prioritized sampling is one kind of solution but not the only one. There are prior works dealing with this issue through filtering data [3] and weighting by uncertainty [4]. Without the comparison with these baselines, it's unclear whether the proposed method is a more effective approach.






[1] Harnessing mixed offline reinforcement learning datasets via trajectory weighting

[2] Awac: Accelerating online reinforcement learning with offline datasets

[3] Bail: Best-action imitation learning for batch deep reinforcement learning.

[4] Uncertainty weighted actor-critic for offline reinforcement learning.

**Questions:**

Please see weaknesses.

---

### Official Review · Reviewer_dnz8 · 2023-11-02

**Soundness:** 2 fair
**Presentation:** 3 good
**Contribution:** 2 fair
**Rating:** 1
**Confidence:** 3

**Summary:**

This study introduces a sampling strategy that assigns priority to certain actions to enhance the efficacy of offline reinforcement learning algorithms with constraints. Traditional offline RL algorithms often restrict the policy to actions observed in the data for each state, which can inadvertently promote the replication of frequent but less optimal actions. The paper presents two novel techniques: ODPR-A (advantage-based) and ODPR-R (return-based). ODPR-A prioritizes state-action pairs for sampling based on the calculated advantage of those pairs within the dataset. Meanwhile, ODPR-R gives priority to state-action pairs from the same sequence by considering the total return of that sequence. These techniques are compatible with a range of algorithms and were tested using five distinct offline RL methods: Behavioral Cloning (BC), Onestep RL, Conservative Q-Learning (CQL), Implicit Q-Learning (IQL), and Twin Delayed DDPG combined with Behavioral Cloning (TD3BC). The results from various offline RL datasets indicate that both ODPR-A and ODPR-R can enhance the algorithms' performance in certain instances.

**Strengths:**

The presentation is straightforward and clear. It's promising to see a comprehensive assessment across five algorithms. The methods introduced do not depend on any specific offline RL algorithm, making them widely applicable. Additionally, the finding that policy evaluation does not need to be prioritized presents a fascinating insight.

**Weaknesses:**

- **Unclear significance:** To strengthen the significance, I suggest the author thinks of experiments answering the question: would the additional complexity (i.e., learning the advantage function, weighting samples) lead to proportional performance gain? Given the experimental results shown in the paper, I see ODPR does better than the baselines, but I'm not convinced that this minor performance gain is worth the added complexity. I suggest showing the results in a more realistic dataset beyond D4RL.

- **Lack of baselines:** The paper's significance is unclear because it lacks a comparison with other resampling/reweighting approaches for improving the dataset. These comparisons, to my opinion, are necessary for positioning this work in the literature and articulating the engineering and practical merits of the methods. For example, AW/RW are much simpler reweighting methods. AW/RW should be compared with the results of Table 2 (Table 7 contains a few, but it would be great to include the results of all five algorithms); otherwise, it's unclear whether ODPR outperforms AW and RW in all D4RL datasets significantly or not. Top-k% sampling is also another simple baseline that the author should consider. Also, in addition, there were several other resampling/reweighting methods that should be considered. Here are a few examples [1, 2, 3, 4], and I hope the author can include the comparison in the rebuttal and upload the experiment scripts to reproduce the results. For [4], I understand ODPR can be a plug-in to [4], but to show the significance of ODPR, it is good to show how much performance gain ODPR can bring to ReDS [4]. If ReDS itself is sufficient to improve the performance, I think the significance of ODPR would be undermined.

- **Lack of comparing with different offline RL parameters:** The author argues that ODPR improves behavior policy and hence improves performance. However, if the problem is about the behavior policy's performance, can one get better performance by tuning the regularization weight (e.g., BC loss weight in TD3+BC, pessimism Q loss in CQL, etc). If so, one doesn't require prioritized sampling. I suggest the author runs a complete hyperparameter sweep over the regularization weight to see if better hyperparameters of offline RL algorithms with uniform sampling can match ODPR.

- **Lack of comparison with behavior policy independent methods:** Theoretically, if the offline RL algorithms are independent of the performance of behavior policy, its performance shouldn't be affected by the data distribution. There have been several well-established methods in this area, like [5, 6]. To strengthen the significance of ODPR, it's good to compare wth them, showing that ODPR has a unique advantage over these existing methods.

- **Rigors of experiment:** In Table 3, the author took the scores directly from AW/RW paper. However, according to [8], the difference in codebase and offline RL hyperparameter choices can make the comparison unfair. To make the empirical results comparable, the author requires implementing AW/RW in the same codebase with ODPR and using the same offline RL hyperparameters.

- **Need better metrics:** The author reports the sum of scores over datasets. However, per [8]'s suggestion, reporting IQM and probability of improvement (PI) are more reliable metrics for reporting aggregated performance. I would like to see if ODPR's performance gain persists in these metrics.

Overall, I want to summarize my review here. I want to see more evidence that supports ODPR should be used in the future for all offline RL problems or some specific datasets that mirror some practical use cases of offline RL. I like the author's approach but more empirical evidence are required to strengthen the value of this paper.

[1] Wu, Yue, et al. "Uncertainty weighted actor-critic for offline reinforcement learning." arXiv preprint arXiv:2105.08140 (2021).

[2] Chen, Xinyue, et al. "Bail: Best-action imitation learning for batch deep reinforcement learning." Advances in Neural Information Processing Systems 33 (2020): 18353-18363.

[3] Xu H, Jiang L, Li J, et al. Offline rl with no ood actions: In-sample learning via implicit value regularization[J]. arXiv preprint arXiv:2303.15810, 2023.

[4] Singh, Anikait, et al. "Offline rl with realistic datasets: Heteroskedasticity and support constraints." arXiv preprint arXiv:2211.01052 (2022).

[5] Kumar, Aviral, et al. "Stabilizing off-policy q-learning via bootstrapping error reduction." Advances in Neural Information Processing Systems 32 (2019).

[6] Cheng, Ching-An, et al. "Adversarially trained actor critic for offline reinforcement learning." International Conference on Machine Learning. PMLR, 2022.

[7] Henderson, Peter, et al. "Deep reinforcement learning that matters." Proceedings of the AAAI conference on artificial intelligence. Vol. 32. No. 1. 2018.

[8] Agarwal, Rishabh, et al. "Deep reinforcement learning at the edge of the statistical precipice." Advances in neural information processing systems 34 (2021): 29304-29320.

**Questions:**

(See Weaknesses)

---

### Meta-Review · Area_Chair_3hez · 2023-12-05

**Metareview:**

This manuscript considers resampling methods in offline RL; however, the technical significance is relatively unclear, the experimental comparison does not consider strong benchmarks, and there is no conceptual basis or analysis for why it exhibits upshots. Therefore, the reviewers are concordant in the assessment that the paper does not meet the bar for acceptance at this time.

**Justification For Why Not Higher Score:**

NA

**Justification For Why Not Lower Score:**

NA

---

### Decision · Program_Chairs · 2024-01-16

Reject